# Chloroplast acquisition without the gene transfer in kleptoplastic sea slugs, *Plakobranchus ocellatus*

**Taro Maeda**[1†]*, **Shunichi Takahashi**[2], **Takao Yoshida**[3], **Shigeru Shimamura**[3], **Yoshihiro Takaki**[3], **Yukiko Nagai**[3], **Atsushi Toyoda**[4], **Yutaka Suzuki**[5], **Asuka Arimoto**[6], **Hisaki Ishii**[7], **Nori Satoh**[8], **Tomoaki Nishiyama**[9], **Mitsuyasu Hasebe**[1,10], **Tadashi Maruyama**[11], **Jun Minagawa**[1,10], **Junichi Obokata**[7,12], **Shuji Shigenobu**[1,10]*

[1]National Institute for Basic Biology, Okazaki, Japan; [2]Sesoko Station, Tropical Biosphere Research Center, University of the Ryukyu, Okinawa, Japan; [3]Japan Agency for Marine-Earth Science and Technology, Yokosuka, Japan; [4]National Institute of Genetics, Shizuoka, Japan; [5]The University of Tokyo, Tokyo, Japan; [6]Marine Biological Laboratory, Hiroshima University, Hiroshima, Japan; [7]Kyoto Prefectural University, Kyoto, Japan; [8]Okinawa Institute of Science and Technology Graduate University, Okinawa, Japan; [9]Research Center for Experimental Modeling of Human Disease, Kanazawa University, Kanazawa, Japan; [10]SOKENDAI, the Graduate University for Advanced Studies, Okazaki, Japan; [11]Kitasato University, Tokyo, Japan; [12]Setsunan Universiy, Hirakata, Japan

**\*For correspondence:**
taromaedaj@gmail.com (TM);
shige@nibb.ac.jp (SS)

**Present address:** [†]Ryukoku University, Otsu, Japan

**Competing interests:** The authors declare that no competing interests exist.

**Abstract** Some sea slugs sequester chloroplasts from algal food in their intestinal cells and photosynthesize for months. This phenomenon, kleptoplasty, poses a question of how the chloroplast retains its activity without the algal nucleus. There have been debates on the horizontal transfer of algal genes to the animal nucleus. To settle the arguments, this study reported the genome of a kleptoplastic sea slug, *Plakobranchus ocellatus*, and found no evidence of photosynthetic genes encoded on the nucleus. Nevertheless, it was confirmed that light illumination prolongs the life of mollusk under starvation. These data presented a paradigm that a complex adaptive trait, as typified by photosynthesis, can be transferred between eukaryotic kingdoms by a unique organelle transmission without nuclear gene transfer. Our phylogenomic analysis showed that genes for proteolysis and immunity undergo gene expansion and are up-regulated in chloroplast-enriched tissue, suggesting that these molluskan genes are involved in the phenotype acquisition without horizontal gene transfer.

## Introduction

Since the Hershey–Chase experiment (*Hershey and Chase, 1952*), which proved that DNA is the material transferred to bacteria in phage infections, horizontal gene transfer (HGT) has been considered essential for cross-species transformation (*Arber, 2014*). Although the prion hypothesis has rekindled the interest in proteins as an element of phenotype propagation (*Crick, 1970*; *Wickner et al., 2015*), HGT is still assumed to be the cause of transformation. For example, in a secondary plastid acquisition scenario in dinoflagellates, (1) a non-phototrophic eukaryote sequesters a unicellular archaeplastid; (2) the endogenous gene transfer to the non-phototrophic eukaryote leads to the shrinkage of the archaeplastidan nuclear DNA (nucDNA); and (3) the archaeplastidan nucleus disappears, and its plastid becomes a secondary plastid in the host (*Reyes-Prieto et al., 2007*).

Chloroplast sequestration in sea slugs has attracted much attention due to the uniqueness of the algae-derived phenotype acquisition. Some species of sacoglossan sea slugs (Mollusca: Gastropoda: Heterobranchia) can photosynthesize using the chloroplasts of their algal food (*Figure 1A and B*; *de Vries and Archibald, 2018*; *Kawaguti, 1965*; *Pierce and Curtis, 2012*; *Rumpho et al., 2011*; *Serôdio et al., 2014*). These sacoglossans ingest species-specific algae and sequester the chloroplasts into their intestinal cells. This phenomenon is called kleptoplasty (*Gilyarov, 1983*; *Pelletreau et al., 2011*). The sequestered chloroplasts (named kleptoplasts) retain their electron microscopic structure (*Fan et al., 2014*; *Kawaguti, 1965*; *Martin et al., 2015*; *Pelletreau et al., 2011*; *Trench, 1969*) and photosynthetic activity (*Cartaxana et al., 2017*; *Christa et al., 2014a*; *Cruz et al., 2015*; *Händeler et al., 2009*; *Taylor, 1968*; *Teugels et al., 2008*; *Wägele, 2001*; *Yamamoto et al., 2009*). The retention period of photosynthesis differs among sacoglossan species (1 to >300 days; *Figure 1B*; *Christa et al., 2015*, *Christa et al., 2014a*, *Christa et al., 2014b*; *Evertsen et al., 2007*; *Laetz and Wägele, 2017*) and development stages and depends on the plastid 'donor' species (*Curtis et al., 2007*; *Laetz and Wägele, 2017*).

The absence of algal nuclei in sacoglossan cells makes kleptoplasty distinct from other symbioses and plastid acquisitions (*de Vries and Archibald, 2018*; *Rauch et al., 2015*). Electron microscopic studies have indicated that the sea slug maintains photosynthetic activity without algal nuclei (*Hirose, 2005*; *Kawaguti, 1965*; *Laetz and Wägele, 2019*; *Martin et al., 2015*; *Pierce and Curtis, 2012*). Because the algal nucleus, rather than the plastids, encodes most photosynthetic proteins, the mechanism to maintain photosynthetic proteins is especially intriguing, given that photosynthetic proteins have a high turnover rate (*de Vries and Archibald, 2018*; *Pelletreau et al., 2011*). Previous polymerase chain reaction (PCR)-based studies have suggested the HGT of algal nucleic photosynthetic genes (e.g., *psbO*) to the nucDNA of the sea slug, *Elysia chlorotica* (*Pierce et al., 1996*; *Pierce et al., 2009*; *Pierce et al., 2007*; *Pierce et al., 2003*; *Rumpho et al., 2008*; *Schwartz et al., 2014*). A genomic study of *E. chlorotica* (N50 = 824 bases) provided no reliable evidence of HGT but predicted that fragmented algal DNA and mRNAs contribute to its kleptoplasty (*Bhattacharya et al., 2013*). *Schwartz et al., 2014* reported in situ hybridization-based evidence for HGT and argued that the previous *E. chlorotica* genome might overlook the algae-derived gene. Although an improved genome of *E. chlorotica* (N50 = 442 kb) was published recently, this study made no mention of the presence or absence of algae-derived genes (*Cai et al., 2019*). The genomic studies of sea slug HGT have been limited to *E. chlorotica*, and the studies have used multiple samples with different genetic backgrounds for genome assembling (*Bhattacharya et al., 2013*; *Cai et al., 2019*). The genetic diversity of sequencing data may have inhibited genome assembling. Although transcriptomic analyses of other sea slug species failed to detect HGT (*Chan et al., 2018*; *Wägele et al., 2011*), transcriptomic data were insufficient to ascertain genomic gene composition (*de Vries et al., 2015*; *Rauch et al., 2015*).

Here, the genome sequences of another sacoglossan species, *Plakobranchus ocellatus* (*Figure 1C–E*), are presented to clarify whether HGT is the primary system underlying kleptoplasty. For more than 70 years, multiple research groups have studied *P. ocellatus* for its long-term (>3 months) ability to retain kleptoplasts (*Christa et al., 2013*; *Evertsen et al., 2007*; *Greve et al., 2017*; *Kawaguti, 1941*; *Trench et al., 1970*; *Wade and Sherwood, 2017*; *Wägele et al., 2011*). However, a recent phylogenetic analysis showed that *P. ocellatus* is a species complex (a set of closely related species; *Figure 1F*; *Christa et al., 2014c*; *Krug et al., 2013*; *Maeda et al., 2012*; *Meyers-Muñoz et al., 2016*; *Yamamoto et al., 2013*). Therefore, it is useful to revisit previous studies on *P. ocellatus*. This study first confirmed the photosynthetic activity and adaptive relevance of kleptoplasty to *P. ocellatus* type black (a species confirmed by *Krug et al., 2013* via molecular phylogenetics; hereafter 'PoB'). The genome sequences of PoB (N50 = 1.45 Mb) and a related species, *Elysia marginata* (N50 = 225 kb), were then constructed. By improving the DNA extraction method, the genome sequences from a single sea slug individual in each species were successfully assembled. The comparative genomic and transcriptomic analyses of these species demonstrate the complete lack of photosynthetic genes in these sea slug genomes and supported an alternative hypothetical kleptoplasty mechanism.

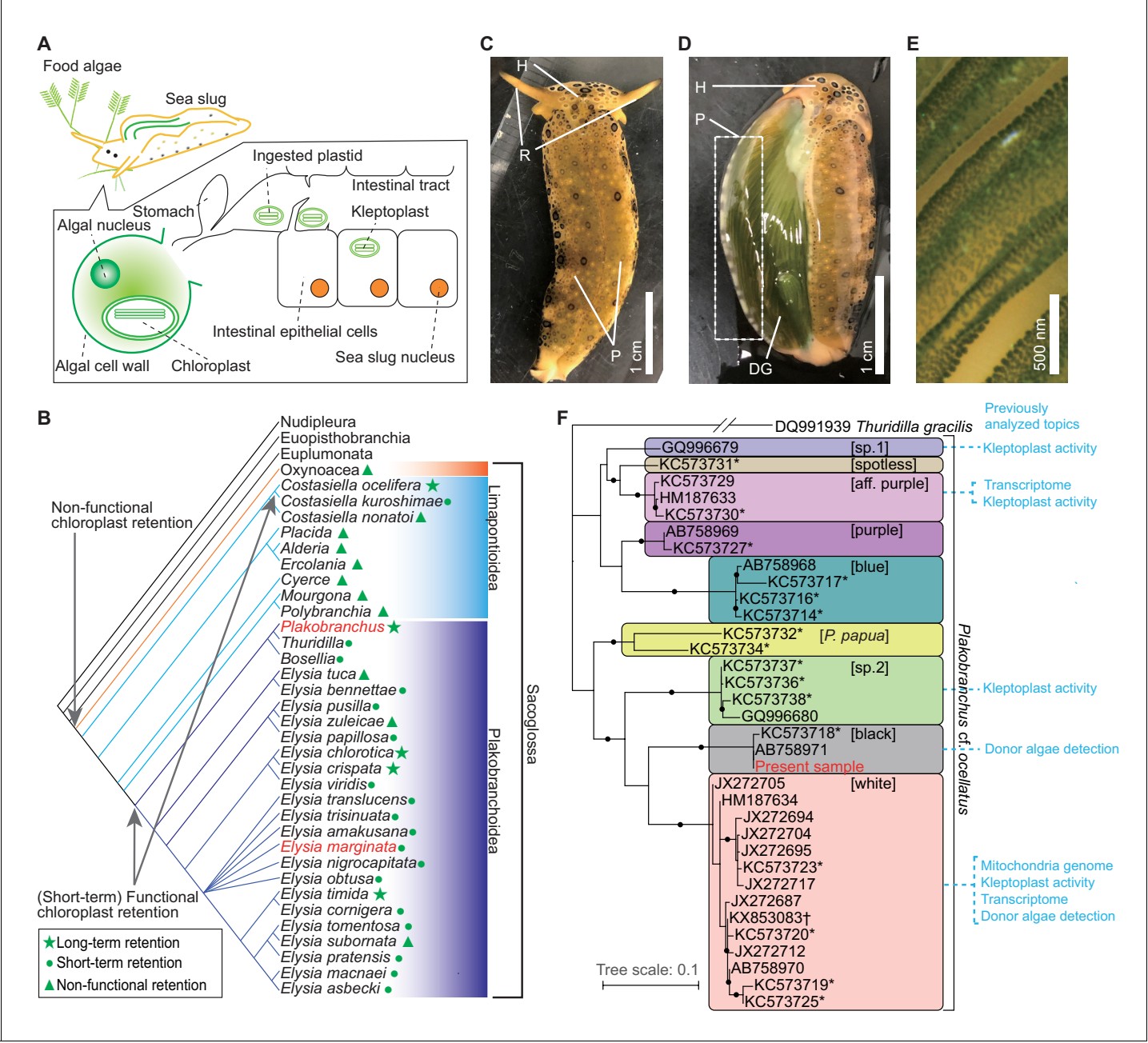

**Figure 1.** Kleptoplasty in sea slugs. (**A**) Process of algal chloroplast retention by a sacoglossan sea slug (*Pierce and Curtis, 2012*). Sacoglossan sea slugs puncture the cell wall of food algae to suck out the protoplasm. The chloroplasts in the protoplasm are transported to the sea slug's intestinal tract, and the intestinal epithelial cells sequester chloroplasts by phagocytosis. The sequestered chloroplasts (kleptoplasts) maintain the photosynthetic activity in the cell for days to months. The sacoglossan cell contains no algal nuclei. Kleptoplast has never been found in germ cells of sea slug. (**B**) Phylogenetic distribution of kleptoplasty in the order Sacoglossa. Phylogenetic analysis showed that a common ancestor of Sacoglossa acquired non-functional chloroplast retention phenomena (without the maintenance of photosynthetic function), and multiple sacoglossan groups subsequently acquired the ability to maintain photosynthetic activity. Phylogenetic tree and kleptoplasty states are simplified from *Christa et al., 2015*. *Christa et al., 2014b* defined functional chloroplast retention for less than 2 weeks as 'short-term retention', and for more than 20 days as 'long-term retention'. Relationships within Heterobranchia are described according to *Zapata et al., 2014*. The red-colored taxa include the species used in the present study (*P. ocellatus* and *E. marginata*). (**C–E**) Photo images of PoB starved for 21 days. (**C**) Dorsal view. H, head; R, rhinophores; P, parapodia (lateral fleshy flat protrusions). Almost always, PoB folded parapodia to the back in nature. (**D**) The same individual of which parapodium was turned inside out (without dissection). The back of the sea slug and inside of the parapodia are green. This coloration is caused by the kleptoplasts in DG, which are visible through the epidermis. (**E**) Magnified view of the inner surface of parapodium. The diagonal green streaks are ridged projections on the inner surface of the parapodium. The cells containing kleptoplasts are visible as green spots. (**F**) Phylogeny of the *P.* cf. *ocellatus* species complex

*Figure 1 continued on next page*

*Figure 1 continued*

based on mitochondrial *cox1* genes (ML tree from 568 nucleotide positions) from INSD and the whole mtDNA sequence. The sequence data for the phylogenetic analysis are listed in *Figure 1—source data 1*. Clade names in square brackets are based on *Krug et al., 2013*. Asterisks mark the genotypes from *Krug et al., 2013*. Study topics analyzed by previous researchers were described within the colored boxes for each cluster. Small black circles indicate nodes supported by a high bootstrap value (i.e., 80–100%). *Thuridilla gracilis* is an outgroup. *Plakobranchus papua* is a recently described species and previously identified as *P. ocellatus* (*Meyers-Muñoz et al., 2016*).

The online version of this article includes the following source data for figure 1:

**Source data 1.** Sequence ID list used for phylogenetic analysis on *Figure 1B*.

## Results

### Does the kleptoplast photosynthesis prolong the life of PoB?

To explore the photosynthetic activity of PoB, three photosynthetic indices were measured: photo-chemical efficiency of kleptoplast photosystem II (PSII), oxygen production rate after starvation for 1–3 months, and effect of illumination on PoB longevity. The Fv/Fm value, which reflects the maximum quantum yield of PSII, was 0.68–0.69 in the 'd38' PoB group (starved for 38 days) and 0.57–0.64 in the 'd109' group (starved for 109–110 days; *Figure 2A*). These values were only slightly lower than those of healthy *Halimeda borneensis*, a kleptoplast donor of PoB (*Maeda et al., 2012*), which showed Fv/Fm values of 0.73–0.76. The donor algae of PoB consisted of at least eight green algal species, and they are closely related to *H. borneensis* (*Maeda et al., 2012*). Although it is not clear whether the Fv/Fm is the same for all donor algae, Fv/Fm is almost identical (~0.83) in healthy terrestrial plants regardless of species (*Maxwell and Johnson, 2000*). Moreover, the values of *H. borneensis* were similar to those of other green algae (e.g., *Chlamydomonas reinhardtii*, Fv/Fm = 0.66–0.75; *Bonente et al., 2012*). Hence, we assumed that Fv/Fm values are similar among the donor species. The Fv/Fm value suggested that PoB kleptoplasts retain a similar photochemical efficiency of PSII to that of the food algae for more than 3 months.

Based on the measurement of oxygen concentrations in seawater, starved PoB individuals ('d38' and 'd109') displayed gross photosynthetic oxygen production (*Figure 2B and C*). The mock examination without the PoB sample indicated no light-dependent increase in oxygen concentrations, i.e., no detectable microalgal photosynthesis in seawater (*Figure 2—figure supplement 1*). The results demonstrated that PoB kleptoplasts retain photosynthetic activity for more than 3 months, consistent with previous studies on *P.* cf. *ocellatus* (*Christa et al., 2014c*; *Evertsen et al., 2007*).

The longevity of starved PoB specimens was then measured under different light conditions. The mean longevity was 156 days under continuous darkness and 195 days under a 12 hr light/12 hr dark cycle (p=0.022; *Figure 2D*), indicating that light exposure significantly prolongs PoB longevity. This observation was consistent with that of *Yamamoto et al., 2013* that the survival rate of PoB after 21 days under starvation is light dependent. Although a study using *P.* cf. *ocellatus* reported that photosynthesis had no positive effect on the survival rate (*Christa et al., 2014c*), our results indicated that this finding does not apply to PoB.

Given the three photosynthetic indices' data, we concluded that the increase in PoB survival days was due to photosynthesis. A previous study using the short-term (retention period of 4–8 days) kleptoplastic sea slug, *Elysia atroviridis*, indicated no positive effect of light illumination on their survival rate (*Akimoto et al., 2014*). These results supported that light exposure does not affect saco-glossan longevity in the absence of kleptoplasts. Although plastid-free PoB may directly indicate that kleptoplasty extends longevity, there is no way to remove kleptoplasts from PoB, except during long-term starvation. PoB feeds on nothing but algae and retains the kleptoplast for 3–5 months. Although future research methods may allow for more experimental analyses of kleptoplast functions, the currently most straightforward idea is that kleptoplast photosynthesis increases the starvation resistance of PoB.

### Do kleptoplasts encode more photosynthetic genes than general plastid?

To reveal the proteins synthesized from kleptoplast DNA (kpDNA), we sequenced the whole kpDNA from PoB and compared the sequences with algal plastid and nuclear genes. Illumina sequencing provided two types of circular kpDNA and one whole mitochondrial DNA (mtDNA) (*Figure 3A*;

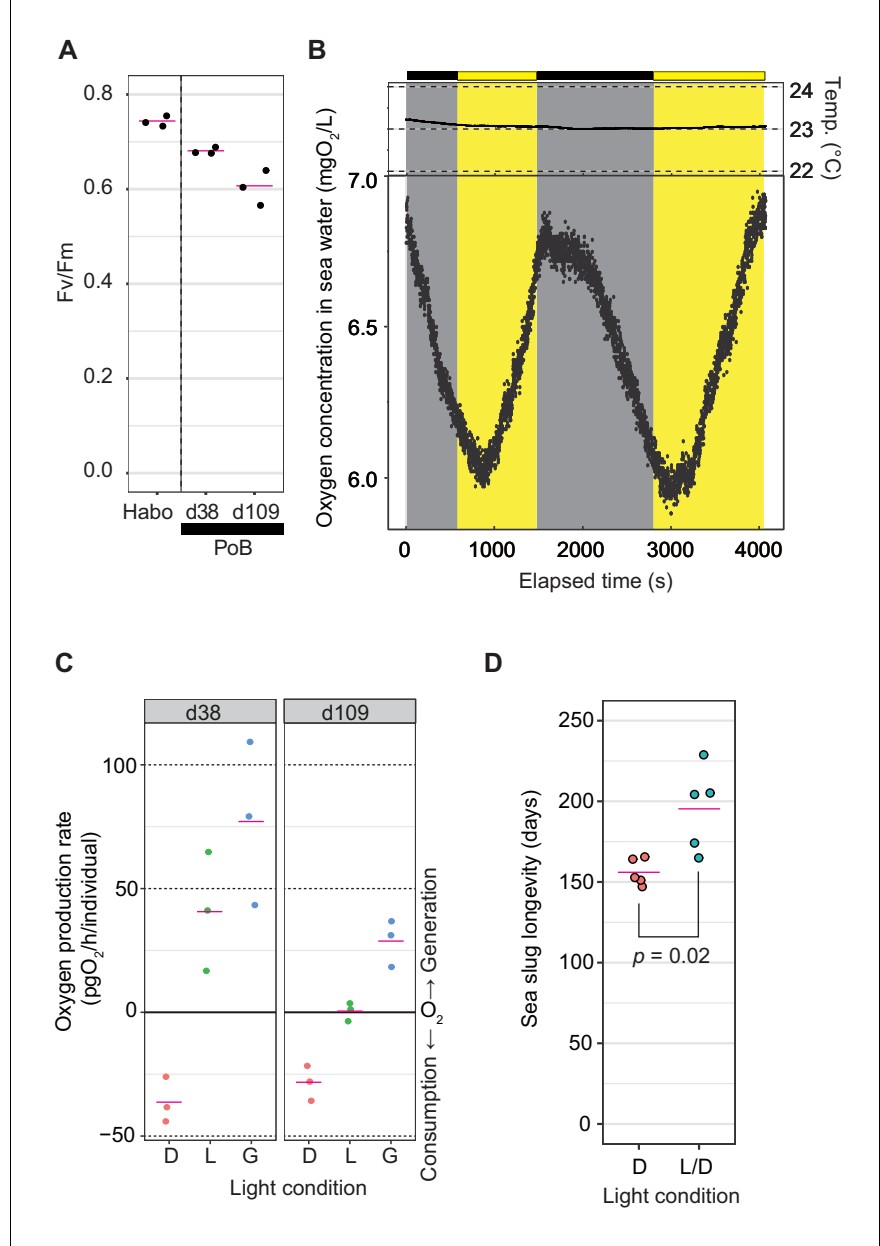

**Figure 2.** Photosynthetic activity of PoB. (**A**) Jitter plot of Fv/Fm values indicating the photochemical efficiency of PSII. Habo, *H. borneensis*; d38, starved PoB for 38 days; d109, starved for 109 to 110 days (12 hr light/12 hr dark cycle; the light phase illumination was 10 μmol photons m$^{-2}$ s$^{-1}$). The magenta line indicates the mean, and the black dot indicates each individual's raw value (n = 3 per group). (**B**) Dynamics of seawater oxygen concentration. To better demonstrate the PoB photosynthesis, a change of oxygen concentration every second was visualized. A PoB individual, one of the d38 samples (ID1), was put into the measurement chamber, and the light conditions were changed in tens of minutes. The measurements in other individuals are visualized in *Figure 2—figure supplement 1*, along with the diagrams of the measuring equipment. Each point represents the oxygen concentration value per second. Gray signifies a dark period, and yellow means an illuminated period (50 μmol photons m$^{-2}$ s$^{-1}$). Temp, water temperature. Although the values fluctuated by the noise due to ambient light and other factors, the oxygen concentration decreased in the dark conditions and increased in the light conditions. When the illumination condition was changed, the changing pattern kept the previous pattern for a few minutes. This discrepancy may reflect the distance between the organism and oxygen sensor and the time for adaptation to brightness by the kleptoplast. (**C**) Jitter plots of PoB oxygen consumption and generation. D, dark conditions; L, light conditions; G, gross rate of light-dependent oxygen generation (L minus D). (**D**) Jitter plots of PoB longevity (n = 5 per group). D, Continuous dark; L/D, 12 hr light/12 hr dark cycle. The p from Welch's two-sample t-test was

*Figure 2 continued on next page*

*Figure 2 continued*

used. A source file of Fv/Fm jitter plot, time course of oxygen concentration, and longevity analysis are available in *Figure 2—source data 1*–3, respectively.

The online version of this article includes the following source data and figure supplement(s) for figure 2:

**Source data 1.** Summarized data of Fv/Fm and oxygen generation activity analysis of *P*.
**Source data 2.** Raw data of oxygen concentration dynamics.
**Source data 3.** Summary of the longevity of analyzed *P. ocellatus* individuals used for *Figure 2D*.
**Figure supplement 1.** Light-dependent oxygen generation by *P. ocellatus*.

*Supplementary file 1*; *Figure 3—figure supplements 1–3*). The mtDNA sequence was almost identical to the previously sequenced *P.* cf. *ocellatus* mtDNA (*Figure 3—figure supplement 3B*; *Greve et al., 2017*). The sequenced kpDNAs corresponded to those of the predominant kleptoplast donors of PoB (*Maeda et al., 2012*), i.e., *Rhipidosiphon lewmanomontiae* (AP014542; hereafter 'kRhip') and *Poropsis* spp. (AP014543; hereafter 'kPoro'; *Figure 3B*; *Figure 3—figure supplement 4*).

To determine whether kpDNA gene repertoires were similar to green algal chloroplast DNAs (cpDNAs), *H. borneensis* cpDNA was sequenced, and 17 whole cpDNA sequences were obtained from public databases (*Figure 3C*; *Figure 3—figure supplements 5* and *6*). PoB kpDNAs contained all the 59 conserved chloroplastic genes in Bryopsidales algae (e.g., *psbA* and *rpoA*), although they lacked four to five of the dispensable genes (i.e., *petL*, *psb30*, *rpl32*, *rpl12*, and *ccs1*; *Figure 3C*).

To test whether kpDNAs contained no additional photosynthetic genes, a dataset of 614 photosynthetic genes was then used (hereafter the 'A614' dataset), which were selected from the algal transcriptomic data and public algal genomic data (*Supplementary files 2* and *3*). A tblastn homology search using A614 obtained no reliable hits (E-value < 0.0001) against the kpDNA sequences, except for the *chlD* gene, which resembled the kpDNA-encoded *chlL* gene (*Figure 3D,E*). A positive control search against an algal nucDNA database (*Caulerpa lentillifera*; https://marinegenomics.oist.jp/umibudo/viewer/info?project_id=55; *Arimoto et al., 2019*) found reliable matches for 93% (575/614) of the queries (*Figure 3D,E*), suggesting that the method has high sensitivity. Hence, it was concluded that kpDNAs lack multiple photosynthetic genes (e.g., *psbO*) as general green algal cpDNAs.

## Are there horizontally transferred algal genes to the PoB nucleic genome?

To determine whether the PoB nucleic genome contains algae-derived genes (i.e., evidence of HGT), we sequenced the nuclear genome of PoB. A search for algae-like sequences in gene models, genomic sequences, and pre-assembled short reads were conducted. The genome assembly contained 927.9 Mbp (99.1% of the estimated genome size; 8647 scaffolds; N50 = 1.45 Mbp; 77,230 gene models; *Supplementary file 4*; *Figure 4—figure supplements 1–3*). Benchmarking Universal Single-Copy Orthologs (BUSCO) analysis using the eukaryota_odb9 dataset showed high coverage (93%) of the eukaryote conserved gene set, indicating that the gene modeling was sufficiently complete to enable HGT searches (*Supplementary file 4*). The constructed gene models were deposited in INSD (PRJDB3267) and FigShare (DOI: 10.6084/m9.figshare.13042016) with annotation data.

Searches of PoB gene models found no evidence of algae-derived HGT. Simple homology searches (blastp) against the RefSeq database found 127 PoB gene models with top hits to Cyanobacteria or eukaryotic algae. Eighty-two of the 127 top hits encoded functionally unknown protein (annotated as 'hypothetical protein' or 'uncharacterized protein'; *Supplementary file 5*). Viridiplantal *Volvox* was the most abundant source of algae of the top-hit sequences (17 genes), and cryptophytan *Guillardia* was the second most abundant source (10 genes). However, the prediction of taxonomical origin using multiple blast-hit results (Last Common Ancestor [LCA] analysis with MEGAN software; *Huson et al., 2007*) denied the algal origin of the genes (*Supplementary file 5*). A blastp search using the A614 dataset, which contains sequences of the potential gene donor (e.g., transcriptomic data of *H. borneensis*), also determined no positive HGT evidence (*Supplementary file 5*).

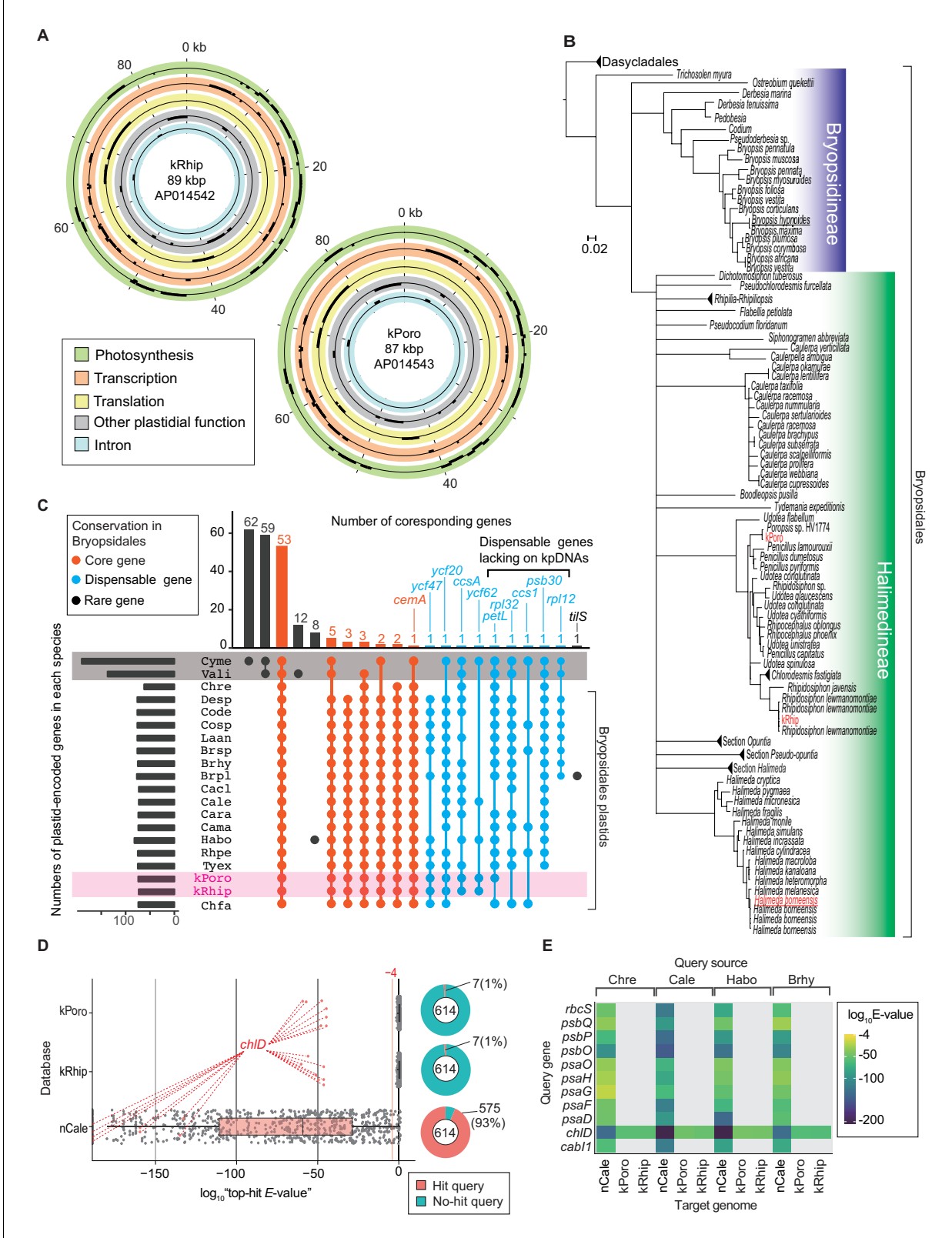

**Figure 3.** Gene composition of PoB kpDNAs. (**A**) Gene map of two kpDNAs from PoB. Gene positions are described in circles colored according to the gene's functional category (see keys in the box). Genes on the outside and inside of each circle are transcribed in the clockwise and anticlockwise directions, respectively (for detailed maps, see *Figure 3—figure supplements 1* and *2*). (**B**) Phylogenetic positions of sequenced kleptoplasts among green algal plastids. The original ML tree (*Figure 3—figure supplement 4*) was created based on *rbcL* genes (457 positions) and converted to the tree. *Figure 3 continued on next page*

*Figure 3 continued*

Red indicates sequenced kpDNA or cpDNA in the present study. Underlines indicate algal species used in RNA-Seq sequencing. (**C**) An UpSet plot of plastid gene composition. Species abbreviations are defined in *Figure 3—source data 1*. The horizontal bar chart indicates the gene numbers in each species. The vertical bar chart indicates the number of genes conserved among the species. Intersect connectors indicate the species composition in a given number of genes (vertical bar chart). Connections corresponding to no gene were omitted. Connectors were colored according to the gene's conservation level in Bryopsidales: Core gene, conserved among all analyzed Bryopsidales species; Dispensable gene, retained more than two Bryopsidales species; Rare gene, determined from a single or no Bryopsidales species. Gray shading indicates non-Viridiplantae algae, and magenta shading indicates PoB kleptoplasts. Cyme (*C. merolae*) and Vali (*V. litorea*) had more than 100 genes that Bryopsidales does not have (e.g., left two vertical bars). (**D**) Box-plots of tblastn results. The vertical axis shows the database searched (kPoro and kRhip, PoB kpDNAs; nCale, nucDNA of *C. lentillifera*). Each dot represents the tblastn result (query is the A614 dataset). Red dots show the result using the *chlD* gene (encoding magnesium-chelatase subunit ChlD) as the query sequence; this sequence is similar to the kleptoplast-encoded *chlL* gene. The right pie chart shows the proportion of queries with hits (E-value < 0.0001). (**E**) Heat map of tblastn results of representative photosynthetic nuclear genes (a subset of data in **D**). The source species of the query sequences are described on the top. The source files of plastid gene composition and tblastn analysis are available in *Figure 3—source data 1* and *2*.

The online version of this article includes the following source data and figure supplement(s) for figure 3:

**Source data 1.** Plastid gene composition.
**Source data 2.** Tblastn analysis of kleptoplast/chloroplast DNA.
**Figure supplement 1.** Gene map of a *P. ocellatus* kleptoplast DNA sequestered from *Rhipidosiphon lewmanomontiae*.
**Figure supplement 2.** Gene map of a *P. ocellatus* kleptoplast DNA sequestered from *Poropsis* spp.
**Figure supplement 3.** Mitochondrial gene map of the *P. ocellatus* type black.
**Figure supplement 4.** Bryopsidales ML tree made by RAxML (a -x 12345 p 12345 -# 300 m GTRGAMMAI) based on 457 positions of *rbcL* genes.
**Figure supplement 5.** Gene map of the *H. borneensis* chloroplast DNA.
**Figure supplement 6.** Gene compositions of 18 algal cpDNAs and *P. ocellatus* kpDNAs.
**Figure supplement 7.** Read coverage depth of the constructed *P. ocellatus* genomic assemblies.
**Figure supplement 8.** Annotation procedure of *H. borneensis* transcripts.
**Figure supplement 9.** Annotation procedure of *B. hypnoides* transcripts.
**Figure supplement 10.** Annotation procedure of *C. lentillifera*.

Using LCA analysis for all PoB gene models, some genes were predicted to originate from species other than Lophotrochozoa. These genes may be due to HGT, but no photosynthesis-related genes were found from them. MEGAN predicted that 20,189 of the PoB genes originated from Lophotrochozoa and its subtaxa. The prediction also assigned 5550 genes to higher adequate taxa as Lophotrochozoa (e.g., Bilateria). MEGAN assigned the 2964 genes to Opisthokonta other than Lophotrochozoa (e.g., Ecdysozoa) and 312 genes derived from proteobacteria as promising horizontally transferred genes. However, these 3276 genes contained no photosynthetic gene. Many of the PoB genes assigned as promising horizontally transferred genes encoded reverse transcriptase. For the remaining 48,215 genes, MEGAN assigned no species. A homology search against the public database annotated 41,203 of the no-taxon-assigned genes as functionally unknown genes (FigShare; DOI: 10.6084/m9.figshare.13042016). Other no-taxon-assigned 7012 genes were not associated with photosynthesis, except one gene (p288c60.92) annotated to 'photosystem I assembly protein ycf3'. However, this annotation to p288c60.92 was not reliable because it seems to be derived from incorrect annotations on public databases. Our reannotation via blastp search against the RefSeq database indicated the similarity of the PoB gene (p288c60.92) to 'XP_013785360, death-associated protein kinase related-like' of a horseshoe crab. The details of the analyses are summarized in *Supplementary file 6*. MEGAN-based analysis, hence, indicated that several non-photosynthetic genes might have originated from proteobacteria or other organisms but provided no evidence of algae-derived HGT.

In gene function assignment with Gene Ontology (GO) terms, no PoB gene model was annotated as a gene relating 'Photosynthesis (GO:0015979),'' although the same method found 72 to 253 photosynthesis-related genes in the five referential algal species (*Figure 4A*). Six PoB genes assigned to the child terms of 'Plastid (GO:0009536)' were found (*Figure 4A*; *Supplementary file 7*). However, an ortholog search with animal and algal genes did not support these six genes' algal origin (*Figure 4—figure supplement 5*). It was considered that the sequence conservation beyond the kingdom caused these pseudo-positives in the GO assignment. The function of these six genes had no relationships to photosynthesis (*Supplementary file 7*); six genes relate proteasome (p2972c65.3), arginine kinase (p197c68.18), DNA-binding response regulator (p234c64.89), chromatin structure

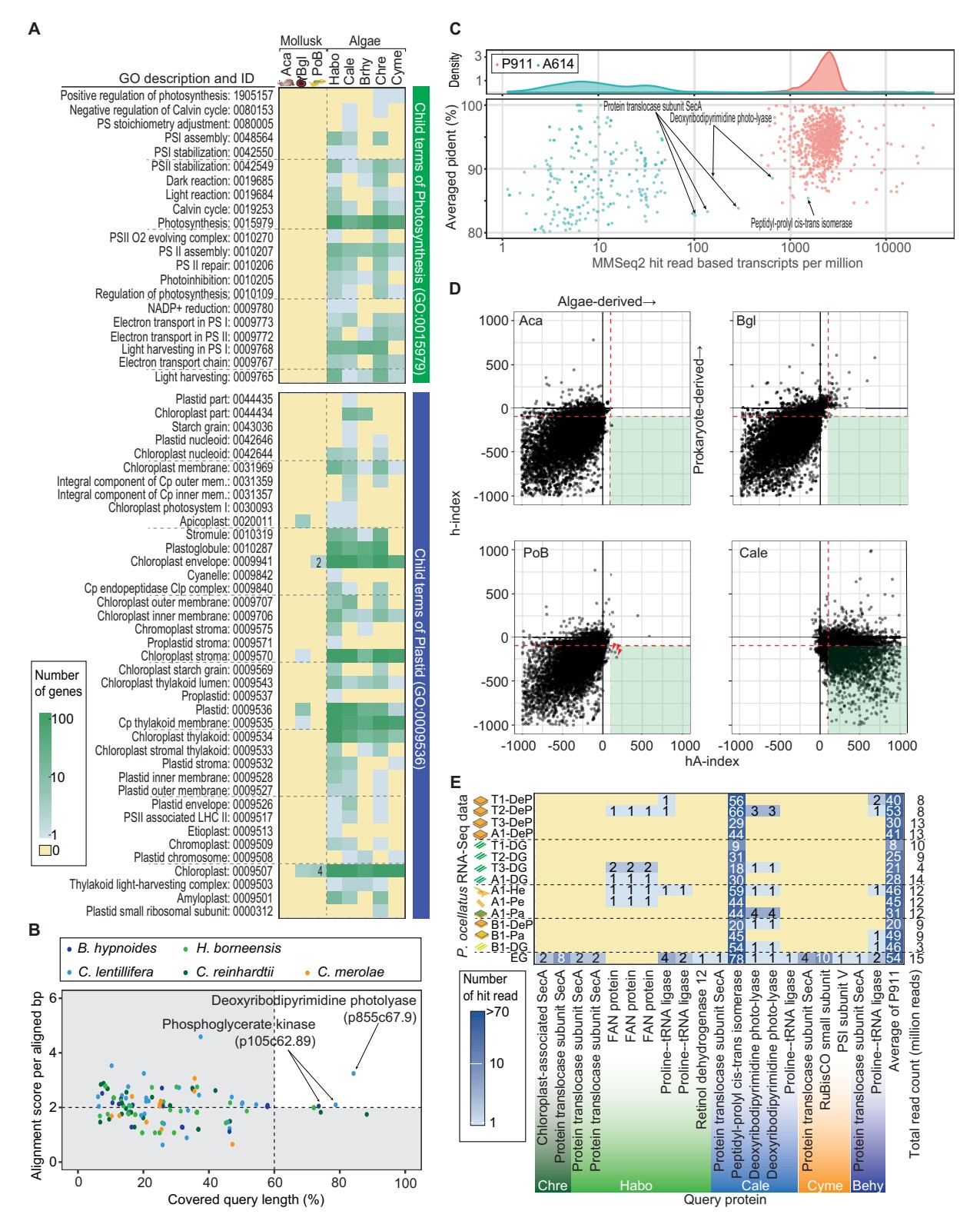

**Figure 4.** Search for horizontally transferred algal genes in the PoB genome. (**A**) Heatmap of the number of genes assigned to photosynthesis- or plastid-related GO terms. The abundance of photosynthesis- or plastid-related genes was compared among PoB, two non-kleptoplastic mollusk species (Aca, *A. californica*; Bgl, *B. glabrata*), and five algal species (see abbreviations in *Figure 3*, *Figure 3—source data 1*). A color scale was visualized in the box on the left. (**B**) Scatter plot of the Exonerate results, the alignment of the A614 gene set (query sequences) to the PoB genome.
*Figure 4 continued on next page*

*Figure 4 continued*

The enlarged view with the tblastn results; *Figure 4—figure supplement 5*. The dot color shows the source algae of each query sequence (see keys in the box). The horizontal axis shows the percentage of the query length aligned to the hit sequences (PoB genome). The vertical axis shows the similarity of the aligned sequences between the query and PoB genome; alignment score (the sum of the substitution matrix scores and the gap penalties) divided by aligned length (bp). Dashed lines are thresholds for a credible query hit (i.e., a hit covering >60% of the query sequence and a normalized Exonerate alignment score of >2). (C) Scatter plot of MMseq2 results for the A614 dataset (algal photosynthetic genes, red) and P911 reference dataset (PoB single-copy genes, blue) used as query sequences against the database of pre-assembled read sets from paired-end DNA libraries of PoB. The top panel shows the probability density distribution of the number of hit reads (normalized with TPM: transcripts per kilobase million; x-axis) versus the averaged 'pident' value (percentage of identical matches) from the hit reads (y-axis). (D) Scatter plot of HGT indices (hA-index versus h-index) for genes in PoB, the two non-kleptoplastic mollusk species (Aca and Bgl), and one algae species (*C. lentillifera* [Cale]). Each dot represents a gene. A high hA-index or h-index value means the possibility of algal or prokaryote origin, respectively. Dashed red lines represent the conventional threshold for HGT (−100 for h-index and 100 for hA-index). Three red arrowheads indicate PoB genes exceeding the thresholds. (E) Heatmap of the results of searches for algae-like RNA fragments in the PoB RNA-Seq data. Pa, parapodium; EG, egg; He, head; Pe, pericardium. The blue gradient indicates the number of RNA-Seq reads assigned as algae-like fragments (see key). The y-axis labels show the RNA-Seq library name and analyzed tissue types. The x-axis labels indicate the query protein; the queries having no corresponding reads were omitted from the figure. For queries using the P911 reference dataset, the mean of the hit-read counts from each library was described. The total number of reads for each library is given on the far right. The source files of homology-based algal gene searching analysis using Exonerate, MMseq2, and blastp are available in raw data *Figure 4—source data 1*.

The online version of this article includes the following source data and figure supplement(s) for figure 4:

**Source data 1.** Algal gene detection from PoB genome.
**Figure supplement 1.** Genome size estimation of *P. ocellatus* type black by flow cytometry.
**Figure supplement 2.** Genome assembling and gene modeling approach for the *P. ocellatus* genome analysis.
**Figure supplement 3.** Annotation procedure for the genomic gene models of *P. ocellatus* type black.
**Figure supplement 4.** ML tree for the candidate HGT genes with algal and molluscan homologous genes.
**Figure supplement 5.** Summary of Exonerate and tblastn results.
**Figure supplement 6.** ML tree for the orthologous group '65' by SonicParanoid.
**Figure supplement 7.** ML tree for the orthologous group '501' by SonicParanoid.
**Figure supplement 8.** ML tree for the orthologous group '456' by SonicParanoid.
**Figure supplement 9.** Sequence similarity between g566.t1 (*C. lentillifera*) and p310c70.15 (*P. ocellatus*).
**Figure supplement 10.** Overview of the sample preparation for *P. ocellatus* type black transcriptomic analysis.
**Figure supplement 11.** Agarose gel electrophoresis image displaying extracted *P. ocellatus* DNA for the nuclear genome sequencing with two fragment size markers.
**Figure supplement 12.** Read depth of the *P. ocellatus* genome assemblies.
**Figure supplement 13.** Dot-plot of the referential kleptoplast DNAs and PoB genomic scaffolds determined as the kleptoplast DNA (kpDNA).
**Figure supplement 14.** Dot plot of the two referential kleptoplast DNAs.
**Figure supplement 15.** Dot plot of the referential mtDNAs and one scaffold determined as mtDNA sequence.

(p466c59.83), mitochondrial inner membrane translocase (p503c65.126), and functionally unknown protein (p45387c41.1).

To confirm that the gene modeling did not overlook a photosynthetic gene, the A614 dataset against the PoB and *C. lentillifera* genome sequences was directly searched using tblastn and Exonerate software (*Slater and Birney, 2005*). Against the *C. lentillifera* genome (positive control), 455 (tblastn) and 450 (Exonerate) hits were found; however, using the same parameters, only 1 (tblastn) and 2 (Exonerate) hits against the PoB genome were detected (*Figure 4B*; *Figure 4—figure supplement 5*). The three PoB loci contain the genes encoding serine/threonine-protein kinase LATS1 (p258757c71.5, *Figure 4—figure supplement 5*), deoxyribodipyrimidine photolyase (p855c67.9, *Figure 4B*), and phosphoglycerate kinase (p105c62.89, *Figure 4B*). A phylogenetic analysis with homologous genes showed the monophyletic relationships of these three genes with molluskan homologs (*Figure 4—figure supplement 6–8*). This result indicated that the three PoB loci contain molluskan genes rather than algae-derived genes. It was thus concluded that the PoB genome assembly contains no algae-derived photosynthetic genes.

To examine whether the genome assembly failed to construct algae-derived regions in the PoB genome, a search for reads resembling photosynthetic genes among the pre-assembled Illumina data was performed. From 1065 million pre-assembled reads, 1698 reads showed similarity against 261 of the A614 dataset queries based on MMseq2 searches. After normalization by query length, the number of matching reads against algal queries was about 100 times lower than that against PoB single-copy genes: the normalized read count (mean ± SD) was 25 ± 105 for the A614 dataset and 2601 ± 2173 for the 905 PoB single-copy genes (p<0.0001, Welch's two-sample t-test;

*Figure 4C*). This massive difference indicated that most algae-like reads come from contaminating microalgae rather than PoB nucDNA. For six algal queries (e.g., peptidyl-prolyl cis-trans isomerase gene), the number of matching reads was comparable between PoB and the A614 dataset (*Figure 4C*; the normalized read count was higher than 100). However, the presence of homologous genes on the PoB genome (e.g., p310c70.15) suggests that the reads were not derived from an algae-derived region but rather resemble molluskan genes. For example, a simple alignment showed that the *C. lentillifera*-derived g566.t1 gene (encoding peptidyl-prolyl cis-trans isomerase) was partially similar to the PoB p310c70.15 gene, and 76% (733/970) of reads that hit against g566.t1 were also hitting against p310c70.15 under the same MMseq2 parameter (*Figure 4—figure supplement 9*). Hence, it was considered that no loss of algal-derived HGT regions occurred in the assembly process.

Changing the focus to HGT of non-photosynthetic algal genes, the indices for prokaryote-derived HGT (h-index; *Boschetti et al., 2012*) and algae-derived HGT (hA-index; see Materials and methods) were calculated for PoB and two non-kleptoplastic gastropods (negative controls; *Supplementary file 8*). Three PoB gene models as potential algae-derived genes were detected (*Figure 4D*); however, two of these encoded a transposon-related protein, and the other encoded an ankyrin repeat protein with a conserved sequence with an animal ortholog (*Figure 4—source data 1*). Furthermore, non-kleptoplastic gastropods (e.g., *Aplysia californica*) had similar numbers of probable HGT genes (*Figure 4D*). Taking these results together, it was concluded that there is no evidence of algae-derived HGT in the PoB genomic data.

The presence of algae-like RNA in PoB was then examined, as a previous study of another sea slug species *E. chlorotica* hypothesized that algae-derived RNA contributes to kleptoplasty (*Bhattacharya et al., 2013*). This previous study used short-read-based blast searches and reverse transcription PCR analyses to detected algal mRNA (e.g., *psbO* mRNA) in multiple adult *E. chlorotica* specimens (no tissue information was provided; *Bhattacharya et al., 2013*). To analyze the algal RNA distribution in PoB, 15 RNA-Seq libraries were constructed from six tissue types (digestive gland [DG], parapodium, DG-exenterated parapodium [DeP], egg, head, and pericardium; *Figure 4—figure supplement 10*), and MMseq2 searches were conducted (*Figure 4C*). Although almost all (594/614) of the A614 dataset queries matched no reads, 19 queries matched 1–10 reads and the *C. lentillifera*-derived g566.t1 query matched more than 10 reads (*Figure 4E*). This high hit rate for g566.t1 (peptidyl-prolyl cis-trans isomerase in *Figure 4E*), however, is due to its high sequence similarity with PoB ortholog, p310c70.15, as mentioned above. A previous anatomical study showed the kleptoplast density in various tissues to be DG > parapodium > DeP = head = pericardium >>> egg (*Hirose, 2005*). Therefore, the amount of algae-like RNA reads did not correlate with the kleptoplast richness among the tissues. Enrichment of algae-like RNA was only found in the egg (*Figure 4E*). The PoB egg is considered a kleptoplast-free stage and is covered by a mucous jelly, which potentially contains environmental microorganisms (*Figure 4—figure supplement 10*). Hence, it was presumed that these RNA fragments are not derived from kleptoplasts but from contaminating microalgae. The searches for algae-like RNA in PoB found no credible evidence of algae-derived RNA transfer and no correlation between algal RNA and kleptoplasty.

## What kind PoB genes are reasonable candidates for kleptoplasty-related genes?

Because the PoB genome was found to be free of algae-derived genes, it was considered that a neo-functionalized molluskan gene might contribute to kleptoplasty. To find candidate kleptoplasty-related molluskan (KRM) genes, we assessed genes that were up-regulated in DG (the primary kleptoplasty location) versus DeP in the RNA-Seq data described above. As a result, 162 DG-up-regulated genes were found (false discovery rate [FDR] < 0.01, triplicate samples; *Figure 5A*; *Figure 5—figure supplements 1* and *2*; *Supplementary file 9*). By conducting GO analysis, the functions of 93 DG-up-regulated genes were identified. These up-regulated genes are enriched for genes involved in proteolysis (GO terms: 'Proteolysis,' 'Aspartic-type endopeptidase activity,' 'Cysteine-type endopeptidase inhibitor activity', and 'Anatomical structure development'), carbohydrate metabolism ('Carbohydrate metabolic process', 'One-carbon metabolic process', 'Cation binding', and 'Regulation of pH'), and immunity ('Defense response'; *Supplementary file 10*). Manual annotation identified the function of 42 of the remaining DG-up-regulated genes. Many of these are also related to proteolysis and immunity: three genes relate to proteolysis (i.e., genes encoding interferon-γ-

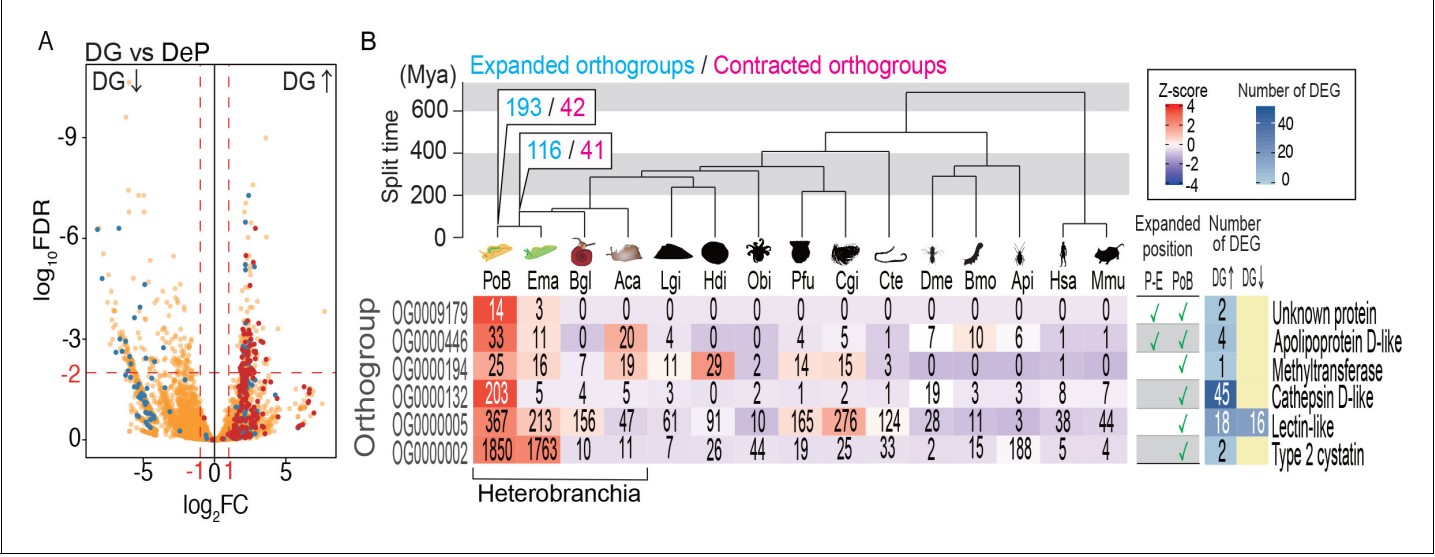

**Figure 5.** Probable KRM genes in PoB. (**A**) Volcano plot comparing gene expressions in DG and DeP tissues of PoB. Red, orthogroup OG0000132 (cathepsin D-like protease genes); blue, orthogroup OG0000005 (lectin-like genes); orange, other orthogroups (for details, see *Figure 5—figure supplement 2*). Up and down arrows signify up- and down-regulation, respectively, in DG. (**B**) Orthogroups expanded on the *P. ocellatus* lineage and containing DG-up-regulated genes. Ema, *E. marginata*; Bgl, *B. glabrata*; Lgi, *Lottia gigantea*; Hdi, *Haliotis discus*; Obi, *Octopus bimaculoides*; Pfu, *Pinctada fucata*; Cgi, *Crassostrea gigas*; Cte, *Capitella teleta*; Dme, *D. melanogaster*; Bmo, *Bombyx mori*; Api, *A. pisum*; Hsa, *Homo sapiens*; Mmu. *Mus musculus*. The phylogenetic tree was scaled to divergence time based on 30 conserved single-copy genes. Mya, million years. The numbers of rapidly expanded (blue) and contracted (magenta) orthogroups on the lineages to PoB are provided at the nodes (for details, see *Figure 5—figure supplements 4* and *5*). Below the tree is information for the six expanded orthogroups that contained DG-up-regulated genes. The left-side heatmap shows the gene numbers (number in boxes) and *Z*-score of gene numbers (color gradient) for each orthogroup. The table shows the expanded/not expanded status of each orthogroup (P–E, *Plakobranchus-Elysia* node; PoB, *Plakobranchus* node). The right-side heat map indicates the number of DEGs between DG and DeP tissue in each orthogroup. Representative gene products are given on the far right. The source files of RNA-Seq analysis and comparative genomic analysis are available in raw data *Figure 5—source data 1* and *2*, respectively.

The online version of this article includes the following source data and figure supplement(s) for figure 5:

**Source data 1.** KRM gene searching using RNA-Seq.
**Source data 2.** KRM gene searching using ortholog analysis.
**Figure supplement 1.** DG-up-regulated genes by the RNA-seq comparison with -DeP samples.
**Figure supplement 2.** Volcano plot of the cross-tissue comparison between DG and DeP samples.
**Figure supplement 3.** The procedure of gene family history analyses.
**Figure supplement 4.** CAFE-based all rapidly expanding/contracting genes on the sacoglossan linage.
**Figure supplement 5.** *Z*-score-based rapidly expanding/contracting genes on the *P. ocellatus* linage (Threshold: Z-score > 2).

inducible lysosomal thiol reductase, replicase polyprotein 1a, and phosphatidylethanolamine-binding protein), and 21 genes contribute to natural immunity (i.e., genes encoding lectin, blood cell aggregation factor, and MAC/perforin domain-containing protein; *Figure 5—figure supplements 1* and *2*). Manual annotation also found four genes encoding apolipoprotein D, which promotes resistance to oxidative stress (*Charron et al., 2008*), and three genes involved in nutrition metabolism (i.e., genes encoding betaine-homocysteine S-methyltransferase 1-like protein, intestinal fatty acid-binding protein, and cell surface hyaluronidase). Because the analyzed slugs were starved for 1 month, it was considered that DG-up-regulated genes contribute to kleptoplasty rather than digestion.

A comparative genomic analysis was then conducted to find orthogroups that expand or contract in size along the metazoan lineage to PoB. The phylogenomic analysis showed that 6 of the 193 orthogroups that underwent gene expansion in this lineage contained DG-up-regulated genes (*Figure 5B*; *Figure 5—figure supplement 3–5*; *Supplementary file 11*). This result supported the notion that these DG-up-regulated genes play a role in kleptoplasty. The most distinctive orthogroup in the six groups was OG0000132, which contained 203 cathepsin D-like genes of PoB. Forty-five of the 203 genes were DG-up-regulated, and Fisher's exact test supported the significant enrichment of DG-up-regulated genes (p<0.0001; *Supplementary file 11*). Other heterobranchian mollusks only had four to five genes belonging to OG0000132 (*Figure 5B*). These gene duplications

in PoB might reduce selection pressure to maintain function via redundancy and promote new function acquirement of the paralogs, as occurs in the well-known neofunctionalization scenario (*Conrad and Antonarakis, 2007*). Significant enrichment of DG-up-regulated genes was also detected in OG0000005 (18 genes) and OG0000446 (four genes; both p<0.0001; *Figure 5B*; *Supplementary file 11*), which contain lectin-like and apolipoprotein D-like genes, respectively. DG-up-regulated genes were also detected in OG0000002, OG0009179, and OG0000194. However, no significant enrichment of these genes was found in these orthogroups (p>0.05). OG0000002 contains DG-up-regulated gene for type two cystatin and various genes with reverse transcriptase domains. This result suggested that the reverse transcriptase domain clustered the various genes as one orthogroup, and the gene number expansion was due to the high self-duplication activity of the retrotransposon in PoB. OG0009179 and OG0000194 contain DG-up-regulated genes of unknown function. From the above results, 67 genes were finally selected as promising targets of study for PoB kleptoplasty: 45 genes for cathepsin D-like proteins, 18 genes for lectin-like proteins, and 4 genes for apolipoprotein D-like protein (*Supplementary file 9*).

## Evolution of candidate KRM genes

For a more detailed analysis of the evolution of KRM genes in sacoglossan lineages, a new draft genome sequence of another sacoglossan sea slug *E. marginata* (previously included in *E. ornata*; *Krug et al., 2013*) was constructed. PoB and *E. marginata* belong to the same family (Plakobranchidae; *Figure 1B*). Both species sequester plastids from Bryopsidales algae; however, the kleptoplast retention time is limited to a few days in *E. marginata* (*Yamamoto et al., 2009*). These features suggested that their common ancestor obtained a mechanism to sequester algal plastids, but *E. marginata* did not develop a system for their long-term retention. Hence, it was considered that comparing gene expansion in these species would clarify the genetic basis of plastid sequestration and long-term retention.

Using the same methods as described for PoB, one complete circular kpDNA, one complete mtDNA, and 790.3 Mbp nucDNA (87.8% of the estimated genome size; 14,149 scaffolds; N50 = 0.23 Mbp; 70,752 genes; *Supplementary files 1*, *4*, *13,* and *14*) were sequenced for *E. marginata*. The constructed gene models covered 89.5% of the BUSCO eukaryota_odb9 gene set (*Supplementary file 4*). No credible photosynthetic gene was detected from *E. marginata* nucDNA (annotation data; DOI: 10.6084/m9.figshare.13042016).

The evolution of representative candidate PoB KRM genes (i.e., cathepsin D-like, apolipoprotein D-like, and lectin-like genes) was then phylogenetically analyzed. In cathepsin D-like genes, sacoglossan (PoB and *E. marginata*) genes formed a specific subgroup in the OG0000132 phylogenetic tree (*Figure 6A*), and gene duplication in OG0000132 seemed to be accelerated along the PoB lineage (203 genes in PoB versus five genes in *E. marginata*; *Figure 5B*). All sacoglossan cathepsin D-like genes belonged to a clade with several other heterobranchian homologs; this clade contained three subclades (α, β, and γ in *Figure 6B*). The basal α-clade contained three *A. californica* genes, one *Biomphalaria glabrata* gene, and three sacoglossan genes. The β- and γ-clades contained sacoglossan genes only and an *A. californica* gene located at the basal position of the β- and γ-clades. Almost all duplicated PoB genes (201/203) belonged to the γ-clade, including one *E. marginata* gene (e8012c40.2). These phylogenetic relationships suggested that the γ-clade has undergone dozens of gene duplication events in the PoB lineage. Interestingly, all DG-up-regulated differentially expressed genes (DEGs) belonged to the γ-clade, and the PoB paralogs that belong to the α- and β-clades showed different expression patterns from the γ-clade paralogs; the gene p609c69.52 (α-clade) was ubiquitously expressed in the examined tissues, and p374c67.53 (β-clade) was expressed only in the egg (*Figure 6B*). The mammalian genes encoding cathepsin D and its analog (cathepsin E) were ubiquitously expressed on various tissue types (*Benes et al., 2008*). Therefore, we considered that (1) the ubiquitously expressed p609c69.52 gene in α-clade is a functional ortholog of the mammalian cathepsin D gene, (2) the p374c67.53 gene in β-clade relates sea slug embryo development, and (3) the γ-clade genes have been acquired with the development of plastid sequestration. To test the positive or relaxed selection on the duplication event, the ratio of substitution rates at non-synonymous (dN) and synonymous (dS) sites ($\omega$ = dN/dS) was used. Longly alignable heterobranchian genes from OG0000132 were used and statistically tested using CodeML and RELAX software (*Supplementary file 12*). The CodeML analysis (branch-site model) determined no significant positive selection on the basal node of the γ-clade (p>0.05, $\chi^2$ test), but RELAX suggested that the

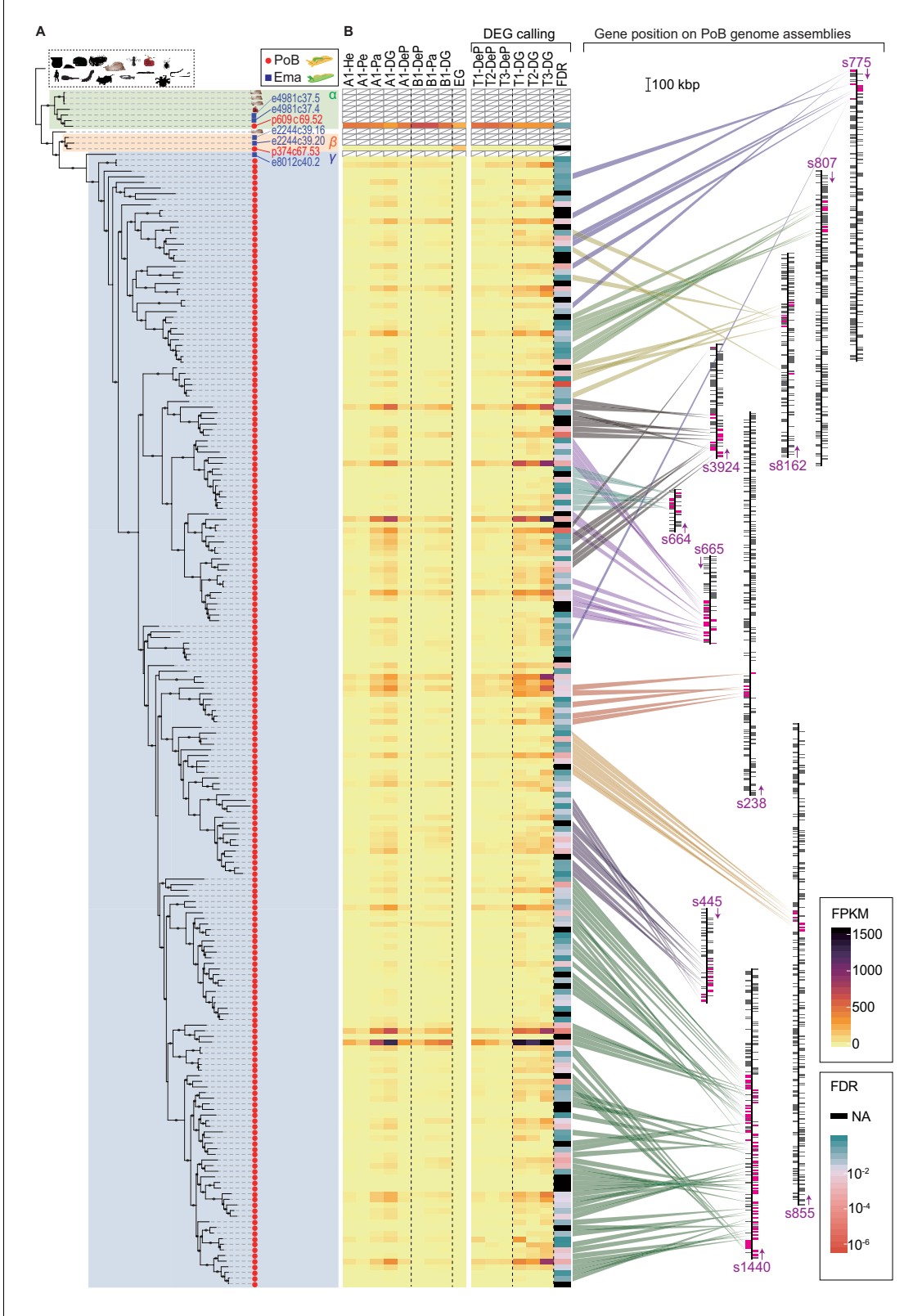

**Figure 6.** Evolutional process of KRM gene candidates of OG0000132. (**A**) ML phylogenetic tree for OG0000132 genes (for details, see *Figure 6—figure supplement 1*). Red circle, PoB gene; blue square, *E. marginata* (Ema) gene. Genes mentioned in the text are indicated with their IDs beside the circles. Pictograms represent non-sacoglossan species defined in *Figure 5—source data 2*. (**B**) Expression of each PoB gene and gene position on the genomic assemblies. The left-side heat plot indicates normalized expression degree (FPKM) of the PoB genes in various tissues; tissue abbreviations

*Figure 6 continued on next page*

*Figure 6 continued*

are defined in *Figure 4*, *Figure 4—figure supplement 10*. The vertical gene order corresponds with the position in (**A**). The white box with a diagonal line means a non-PoB gene (no expression data is available). Tissue samples for the FDR calculation (DG versus DeP) were enclosed with square brackets. Color scales for FPKM and FDR are visualized on the right-side boxes. The FDR's black color indicates that the value could not be calculated (NA) due to its undetectable gene expression. The right panel shows the genomic positions of the genes. The bands' color indicates the correlation between the gene and position, purple text and arrows indicate the scaffold ID and direction, and colored boxes on the scaffolds mean the genes' positions (magenta, OG0000132 gene; gray box, other protein-encoding genes). Scaffolds having less than five OG0000132 genes were omitted from the figure.

The online version of this article includes the following source data and figure supplement(s) for figure 6:

**Source data 1.** Outputs of a prediction of protein subcellular localization via DeepLoc.
**Figure supplement 1.** Enlarged view of gene tree and annotation data for all member of OG0000132 (cathepsin D-like).
**Figure supplement 2.** Positions of genes belonging to OG0000132 on the *E. marginata* genome.
**Figure supplement 3.** Gene tree and annotation data for all genes of OG0000446 (apolipoprotein D-like).
**Figure supplement 4.** Gene positions of OG0000446 on *E. marginata* genome.
**Figure supplement 5.** Gene tree for the member of OG0000005 (lectin-like).

γ-clade evolved under relaxed selection relative to the rest branches (K = 0.44, p<0.001). The cathepsin D-like genes formed multiple tandem repeat structures in the PoB genome, although other Heterobranchia had no tandem repeat (*Figure 6B*; *Figure 6—figure supplements 1* and *2*). In *E. marginata*, the gene e8012c40.2 located at the basal position of the γ-clade had no repeat structure, although two genes from the α-clade (e4981c37.5 and e4981c37.4) and two from the β-clade (e2244c39.16 and e2244c39.20) made two tandem repeats (*Figure 6—figure supplements 1* and *2*). The tree and repeat structure suggested that the γ-clade was separated from the β-clade as a single-copy gene in the common ancestor of PoB and *E. marginata* and duplicated in the PoB lineage (*Figure 6B*). The revealed genomic structure indicated that the duplicates were not due to whole-genome duplication but rather a combination of several subgenomic duplication events: the dispersed duplicates are likely due to replicative transposition by a transposable element, and the tandem repeats are likely due to a local event (e.g., unequal crossover).

The gene duplication in OG0000446 seems to have happened in the PoB lineage and at the node between PoB and *E. marginata* (*Figure 5B*; *Figure 6—figure supplements 3* and *4*). The sacoglossan genes were duplicated in a monophyletic clade (clade I) only, and all DG-up-regulated DEGs were contained in clade I. CodeML found no significant positive selection on the basal node of clade Ia (p>0.05), but RELAX suggested evolution under relaxed selection relative to the rest branches (K = 0.69; p=0.003; *Supplementary file 12*). On the other subclades, no significant positive selection or relaxed selection was found (*Supplementary file 12*). All DG-up-regulated DEGs were contained in clade I. It was hypothesized that duplication on the common lineage relates to plastid sequestration, and the PoB-specific duplication events contribute to long-term kleptoplasty.

In OG0000005 (lectin-like gene group), the gene counts were comparable between PoB (367 genes) and *E. marginata* (213 genes; *Figure 5B*); however, the phylogenetic tree suggests that this similarity is due to different gene duplication events in each species. One-by-one orthologous gene pairs were rare between PoB and *E. marginata* (only 14 were detected), and many of the other homologs formed species-specific subclades (*Figure 6—figure supplement 5*). Lectins are carbohydrate-binding molecules that mediate the attachment of bacteria and viruses to their intended targets (*Lis and Sharon, 1998*; *Yamasaki et al., 2008*). The observed gene expansions of lectin-like genes in each of these sea slugs may widen their targets and complicate the natural immune system to distinguish the kleptoplast from other antigens. Four clades having expanded homologs on the sacoglossan lineage were detected (clades A–D in *Figure 6—figure supplement 5*). These clades contained all 34 DEGs between DG and DeP tissues. The DG-up-regulated and DG-down-regulated genes are located on different clades, except for one DG-up-regulated gene (p3334c67.98) in clade A. These results indicated that the determined DEGs have different expression patterns depending on the clade.

## Discussion

Here, it was demonstrated that (1) kleptoplast photosynthesis extends the lifetime of PoB under starvation, (2) the PoB genome encodes no algal nucleus-derived genes, and (3) PoB individuals up-regulate genes for carbohydrate metabolism, proteolysis, and immune response in their DG. Candidate KRM genes, i.e., apolipoprotein D-like, cathepsin D-like, and lectin-like genes, were also identified using a combination of genomic and transcriptomic analysis. Data present a paradigm of kleptoplasty in which PoB obtains the adaptive photosynthesis trait by a HGT-independent phenotype acquisition.

The PoB genomic sequence clarified the gene repertory of its kpDNA, mtDNA, and nucDNA. The whole kpDNA sequence indicated that PoB kleptoplasts could produce some proteins involved in photosynthesis (e.g., PsbA, a core protein in PSII; *Figure 3*) if gene expression machinery is sufficiently active, as reported in *E. chlorotica* (*Green et al., 2000*; *Pierce et al., 2007*). The absence of core photosynthetic genes in the PoB genome was then demonstrated. For instance, no genes encoding PsaF of photosystem I, PsbO of PSII, or RbcS of Rubisco in kpDNAs, mtDNA, or nucDNA (*Figures 3* and *4*) were detected despite the queries (e.g., A612 dataset; *Supplementary file 3*) containing multiple algal orthologs of these genes; these gene products are essential for photosynthesis in various plants and algae and located on their nuclear genome, not plastid genome (*Farah et al., 1995*; *Izumi et al., 2012*; *Pigolev et al., 2009*). This result means that PoB can perform photosynthesis (*Figure 2*) without de novo synthesis of these gene products. The absence of algae-derived HGT is consistent with previous transcriptomic analyses of *P.* cf. *ocellatus* (*Wägele et al., 2011*) and other sacoglossan species (*de Vries et al., 2015*; *Han et al., 2015*; *Wägele et al., 2011*). A previous genome study of *E. chlorotica* predicted that fragmented algal DNA and mRNAs contribute to its kleptoplasty (*Bhattacharya et al., 2013*), and fluorescence in situ hybridization study detected algal gene signals on *E. chlorotica* chromosomes (*Schwartz et al., 2014*). Although our manual check of gene annotation found no photosynthesis-related gene from a more complete version of the *E. chlorotica* nuclear genome sequence, *Cai et al., 2019* provided no discussion about HGT. After all, the PoB results proposed that algal DNA and/or RNA is not an absolute requirement for kleptoplast photosynthesis.

The combination of genomics and transcriptomics suggests that the maintenance of algae-derived protein activity is the most probable mechanism for retaining PoB photosynthesis. Because of the limited longevity of the photosynthetic proteins in algal cells and/or in vitro (*Roberts et al., 2013*), previous studies have discussed the elongation of algal protein lifespans via protective sea slug proteins as an alternative hypothesis to HGT (*de Vries and Archibald, 2018*; *Serôdio et al., 2014*). This study shows three types of potential KRM genes: apolipoprotein D-like, cathepsin D-like, and lectin-like genes. Some of these KRM genes may encode proteins that protect algae-derived proteins. Although further studies are needed to elucidate their functional roles in detail, the obtained results are consistent with HGT's alternative hypothesis, algal protein retention.

Previous RNA-Seq studies of *Elysia timida* and *E. chlorotica* found up-regulation of superoxide dismutase (SOD) genes in response to photostress and postulated that SOD protects algal proteins in the kleptoplasts from oxidative damage (*Chan et al., 2018*; *de Vries et al., 2015*). No significant up-regulation of the SOD gene in PoB DG was found (*Figure 5—figure supplement 1*). However, there was up-regulation of apolipoprotein D-like genes and the expansion of these genes in PoB lineages (*Figure 5*). Apolipoprotein D, a lipid antioxidant, confers resistance to oxidative stress in higher plants and animal brains (*Bishop et al., 2010*; *Charron et al., 2008*). Ortholog analysis of the new *E. chlorotica* data found no gene number expansion of these homologs in *E. chlorotica*, i.e., only three apolipoprotein D-like homologs (*Supplementary file 15*). Although the retention process details may differ among species and abiotic conditions, it is attractive to speculate that oxidative stress resilience is of major importance for kleptoplasty in multiple sacoglossan species.

Previous kleptoplasty studies have not focused on cathepsin D-like and lectin-like genes. The involvement of the mechanisms of these proteins in kleptoplasty remains to be solved. Cathepsin D degrades intracellular proteins and contributes to the degradation of damaged mitochondria (*Benes et al., 2008*). In general, damaged photosynthetic proteins generate abundant ROS, promoting further protein damage. In PoB, cathepsin D-like protease may degrade damaged proteins and prevent further protein inactivation. Lectin is involved in self-recognition in natural immunity (*Geijtenbeek and Gringhuis, 2009*; *Worthley et al., 2005*). The diverse lectins expressed in PoB

DG tissue may bind the antigens of algae-derived molecules, mediate the detection of non-self-proteins and/or saccharides, and lead to the selective degradation and retention of algae-derived proteins and organelles.

These genomic data indicate that a proteomic analysis of kleptoplasts is warranted. A previous isotopic study indicated that *E. chlorotica* kleptoplasts contain function-unknown proteins transported from the sea slug cytoplasm (*Pierce et al., 1996*). Although several algal photosynthetic proteins have been immunoassayed in kleptoplasts, animal nuclear-encoded proteins have been out of the target (*Green et al., 2000*; *Pierce et al., 1996*). The in silico study found no typical chloroplast localization signal in PoB KRM genes (*Figure 6—figure supplements 1* and *3*). However, the genomic data will help identify kleptoplast-localized sea slug proteins by peptide mass fingerprinting.

This study provides the first genomic evidence of photosynthesis acquisition without horizontal DNA or RNA transfer. Previous studies have demonstrated that DNA is the core material for heredity (*Hershey and Chase, 1952*; *Watson and Crick, 1953*) and assumed that horizontal DNA transfer causes cross-species phenotype acquisition (*Acuña et al., 2012*; *Anderson, 1970*; *Luis, 2014*; *Dehal et al., 2002*). However, our studies indicate that PoB gains adaptive photosynthetic activity without acquiring any of the many algal nucleic genes involved in photosynthesis. This is evidence of a phenotype (and organelle) acquisition, although temporally, from the donor without DNA or RNA transfer. Recent studies of shipworms (wood-feeding mollusks in the family Teredinidae) showed that they utilize several symbiont-derived proteins for their food digestion (*O'Connor et al., 2014*). Golden sweeper fish (*Parapranthus ransonneti*) gain the luciferase for their bioluminescence from ostracod prey, suggesting phenotype acquisition via the sequestration of a non-self-protein (kleptoprotein; *Bessho-Uehara et al., 2020*). However, these two examples of HGT-independent phenotype acquisition are limited to the transfer of simple phenotypes that depend on just a few enzymes. In contrast, the well-known complexity of photosynthesis suggests that sea slug kleptoplasty depends on DNA/RNA-independent phenotype acquisition of complex pathways requiring multiple enzymes (e.g., entire photosystems and the Calvin cycle). It is attractive to speculate that other symbiont-derived organelles (e.g., mitochondria) and obligate endosymbiotic bacteria and protozoan kleptoplasts (e.g., in *Dinophysis acuminata*; *Hackett et al., 2003*) can obtain traits derived from another species via a HGT-independent system. Although several organisms have multiple HGT-derived functional genes, it is still unclear how the organism evolutionarily obtained the appropriate expression control system of the non-self-gene (*Sasakura et al., 2016*). These PoB data suggested that the transfer of complex adaptive phenotypes sometimes precede the gene transfer from the donor species, having the potential to explain the process of cross-species development of complex phenotypes. Some organisms might evolutionarily obtain the HGT-derived genes and appropriate control system of the gene expression after the transfer of phenotype. Our finding of HGT-independent complex phenotype acquisition may open new viewpoints on cross-species evolutionary interaction.

## Materials and methods

### Samplings of sea slugs and algae

Samples were collected from southwestern Japan. Specifically, PoB and *H. borneensis* were collected on Okinawa Island shores, and *E. marginata* was collected from Kinkowan Bay. Regarding *B. hypnoides*, a cultivated thallus was initially collected from Kinkowan Bay and used in the authors' laboratory for several years. The collected samples of PoB and *E. marginata* in seawater were transported to the laboratories at the National Institute for Basic Biology (NIBB) and Kyoto Prefectural University, respectively, under dark conditions within 2 days. The samples were then acclimated in an aquarium filled with artificial seawater (REI-SEA Marine II; Iwaki, Japan) at 24°C.

### Photosynthetic activity of sea slugs and algae

The photosynthetic activity of PoB was measured after 38, 109, or 110 days of incubation under a 12 hr light/12 hr dark cycle without food. During the light phase, the photosynthetic photon flux density was 10 µmol photons $m^{-2}$ $s^{-1}$ (LI-250A Light Meter with LI-193 Underwater Spherical Quantum Sensor; LI-COR, Lincoln, NE). The seawater was not changed during the incubation period, except to adjust the salinity using distilled water. Photosynthetic activity indices (oxygen generation rate and

PAM fluorometry) were measured using oxygen sensor spots (Witrox 4; Loligo Systems, Tjele, Denmark) and PAM-2500 (Walz, Effeltrich, Germany), respectively (*Figure 2—figure supplement 1B–F*). *H. borneensis*, which was easily collected and kept in the laboratory, was used as a reference for comparison in the PAM analysis. This species is a donor alga of PoB kleptoplasts and closely related species with the major donors of PoB, *R. lewmanomontiae*, and *Poropsis* spp. These two major donors firmly adhered to the rock, were tangled up in other algal species, and were difficult to be collected.

The oxygen sensor spots were affixed to the inside of a glass respirometry chamber. Before performing the measurements, the system was calibrated using sodium sulfite (0% $O_2$ saturation) and fully $O_2$-saturated seawater (100% $O_2$ saturation). A sea slug was placed into a respirometry chamber filled with fully $O_2$-saturated filtered artificial seawater (7 ml). The top of the chamber was closed with a glass slide. All visible bubbles were removed from the chamber. The chamber was maintained at a constant temperature (23–24°C) using a water jacket attached to a temperature-controlled water flow. The Witrox temperature probe for calibration was immersed in the water jacket (*Figure 2—figure supplement 1B–F*).

The oxygen concentration was measured sequentially under changing light conditions. The percent $O_2$ saturation was monitored continuously and recorded using AutoResp software (Loligo Systems) for 10 min after the respirometry chamber acclimation period (10 min). The oxygen consumption rate by respiration was measured under dark conditions. Next, the respirometry chambers were exposed to the red LED light (800 μmol photons m$^{-2}$ s$^{-1}$) to measure the change in the oxygen concentration under light conditions due to the balance between the photosynthesis and respiration rates. The chambers were illuminated from the sides because a mounted LED light increased the oxygen meter's noise. The light direction to the sea slug was inconsistent because the slug kept moving around the chamber during the measurement period. In sea slugs, the rate of photosynthesis was unaffected by the direction of illumination because the rate of $O_2$ generation rate under a particular constant illumination did not change regardless of the sea slug's position in the chamber. The percent $O_2$ saturation was measured for 10 min. One blank (i.e., without sea slug) condition was run as a negative control to account for background biological activity in seawater. AutoResp software was used to convert the percent saturation to an oxygen concentration ([$O_2$], mg $O_2$ l$^{-1}$) based on the rate of change in the percent $O_2$ saturation, water temperature, and barometric pressure (fixed at 1013 Pa). A regression analysis was performed using the 'lm' function in R version 3.5.2 (tidyverse 1.2.1 package) to calculate the changing oxygen concentration rate under dark and light conditions. A gross oxygen generation rate was then obtained by photosynthesis (oL + oD = oG; oL, oxygen production rate under light conditions; oD, oxygen consumption rate under dark conditions; oG, gross oxygen generation by photosynthesis).

During PAM fluorometry analysis, each sea slug was caged in a single well of a 12-well cell culture plate (Corning, Corning, NY) after adaptation to the dark for 15 min. To ensure reproducibility, the sea slug was caged upside down (i.e., the ventral surface was brought to the upside), the animal was softly squeezed with a plastic sponge, and the PAM light probe was connected to the plate from underneath the well. Consequently, the samples could not move during the measurement, and the PAM light probe always measured the fluorescence of the dorsal surface. The maximal quantum yield, Fv/Fm, was determined by a saturation pulse of >8000 μmol photons m$^{-2}$ s$^{-1}$ and a measurement light of 0.2 μmol photons m$^{-2}$ s$^{-1}$.

## Effect of light conditions on *P. ocellatus* longevity

The longevity of PoB was measured using a modified medaka (Japanese rice fish) housing rack system (Iwaki, Japan). For the longevity measurement, different individuals from the photosynthetic activity measurement were used. The longevity was investigated from the samples used in the above-mentioned photosynthetic activity measurement. Using centrally filtered systems, the water tank rack maintained consistent water conditions (e.g., temperature and mineral concentrations) among the incubation chambers (subtanks) and enabled a focus on the effect of light conditions. After acclimating the collected sea slugs under the same conditions for 1 week in an aquarium, the organisms were incubated separately for 8 months under different light conditions (continuous dark and 12 hr light/12 hr dark cycle). The sea slugs' conditions were measured daily, and death was defined when a sea slug remained motionless for 30 s after stimulation (i.e., touching with plastic bar).

## Sequencing of *H. borneensis* cpDNA

A combination of pyrosequencing and Sanger sequencing was used to evaluate *H. borneensis* cpDNA. Collected *H. borneensis* thalli were washed with tap water to remove the attached organisms. The cleaned thalli (27 g) were frozen in liquid nitrogen, ground with a T-10 basic homogenizer (IKA, Germany) to a fine powder, suspended in 15 ml AP1 buffer from a DNeasy Plant Mini Kit (Qiagen, Hilden, Germany), and centrifuged (500 × *g*, 1 min) to remove the calcareous parts. Total DNA was purified from the supernatant according to the protocol supplied with the DNeasy Plant Mini Kit. The resulting DNA yield (76.8 µg) was measured using a dsDNA HS Assay Qubit Starter Kit (Thermo, Waltham, MA) and used to prepare a single-fragment library for pyrosequencing. The Pyrosequencer GS-FLX Titanium Platform (Roche, Germany) was used to generate 23.04 Mb of total singleton reads (68032 reads; average read length, 368 bp). After filtering low-quality reads, the remaining 21865 reads (~8 Mb) were submitted for assembly by Newbler (Roche). Of the 6309 obtained contigs (N50 = 663 bp), the cpDNA sequences were identified by blastx searches (version 2.2.28) against the protein coding sequences of cpDNA from the chlorophyte alga *B. hypnoides* (NC_013359). The gaps between the four identified contigs (55551, 21264, 8076, and 3435 bp) and ambiguous sites in the contigs were amplified by PCR using inverse primers. The PCR products were sequenced by primer walking and Sanger sequencing with Takara LA Taq (Takara, Japan), a dGTP BigDye Terminator Cycle Sequencing FS Ready Reaction Kit (Thermo), and an ABI PRISM 3130xl DNA Sequencer (Thermo). Regions that could not be read by direct sequencing were amplified using specific primers, cloned with a TOPO TA cloning kit (Thermo), and sequenced with plasmid-specific primers. Complete cpDNA sequences were obtained by assembling the GS-FLX contigs and reads generated by Sanger sequencing using Sequencher version 4.10 (Gene Codes Corporation, Ann Arbor, MI). All open reading frames > 100 bp were annotated using a blastx search version 2.2.31+ against the non-redundant protein sequences (nr) database in GenBank and a tblastx search of chloroplast genes from other algae (*B. hypnoides*, *C. reinhardtii*, *Vaucheria litorea*, and *H. borneensis*). Introns were detected using RNAweasel (*Gautheret and Lambert, 2001*), Rfam (*Kalvari et al., 2018*), and Mfold (*Zuker, 2003*). The splicing sites were confirmed via alignment with orthologous genes from other green algal cpDNAs. MAFFT version 7.127b (*Katoh and Standley, 2013*) was used to perform the alignment. The origins of bacteria-like proteins were explored using a blastx search against the nr database and a phylogenetic analysis with blast-hit sequences. MAFFT (*Katoh and Standley, 2013*) was used for alignment, Trimal version 1.4 (*Capella-Gutiérrez et al., 2009*) was used for trimming, and RaxML version 8.2.4 (*Stamatakis, 2014*) was used for phylogenetic tree construction. The resulting gene maps were visualized using Circos version 0.69–2.

## Sequencing of kpDNAs from *P. ocellatus*

Kleptoplast sequences were generated using an Illumina system. Total DNA was extracted from DG (kleptoplast-rich tissue) and parapodium (including DG, kleptoplast-less muscle, and reproductive systems) of a single PoB individual using a CTAB-based method (*Murray and Thompson, 1980*). Two Illumina libraries with 180 and 500 bp insertions were constructed from each DNA pool (DRR063261, DRR063262, DRR063263, and DRR063264). An S220-focused ultrasonicator (Covaris, Woburn, MA), Pippin Prep (Sage Science, Beverly, MA), and a TruSeq DNA Sample Prep Kit (Illumina, San Diego, CA) were used for DNA fragmentation, size selection, and library construction, respectively. Libraries were sequenced (101 bp from each end) on a HiSeq 2500 platform (Illumina). A total of 42,206,037 raw reads (8.53 Gb) were obtained. After filtering the low-quality and adapter sequences, the remaining 5.21 Gb of sequences were used for assembly. Paired sequences from 180 bp libraries were combined into overlapping extended contigs using FLASH version 1.2.9 (*Magoč and Salzberg, 2011*) with the default settings. An input of 14,867,401 paired-end sequences and FLASH were used to construct 13,627,554 contigs (101–192 bp). The joined fragments and filtered paired sequences from 500 bp libraries were assembled using Velvet assembler version 1.2.07 (*Zerbino and Birney, 2008*), with parameters that were optimized based on the nucDNA and kpDNA sequence coverage depths; the estimated nucDNA depth was approximately 30× based on the k-mer analysis, and the predicted kpDNA depth was 272× based on the read mapping to previously obtained kleptoplast *rbcL* sequences (AB619313; 1195 bp) using Bowtie2 version 2.0.0 (*Langmead and Salzberg, 2012*). After several tests to tune the Velvet parameters, the best assembly was achieved with a k-mer of 83 and exp_cov of 50. The resulting assembly comprised 1537

scaffolds (>2000 bp) containing 4,743,113 bp (N50 = 2830 bp). Two kpDNAs (AP014542 and AP014543) from this assembly were then identified based on the sequence similarity with *H. borneensis* cpDNA and a mapping back analysis. blastx version 2.2.31 assigned bit scores > 1000 to the two scaffolds (AP014542 = 1382, AP014543 = 1373, database = coding sequences in the obtained *H. borneensis* cpDNA, and query = all constructed scaffolds). Mapping back, which was performed using BWA version 0.7.15-r1140, showed that the coverage depths of AP014542 and AP014543 correlated with the relative abundance of kleptoplasts; the read's average coverage depth was increased by twofold to fourfold in a library from kleptoplast-rich tissue (DRR063263) relative to a library from kleptoplast-poor tissue (DRR063261). The same degree of change was never observed in other scaffolds (*Figure 3—figure supplement 7*).

The two kleptoplast sequences were annotated and visualized using the same method described for *H. borneensis* cpDNA. The phylogenetic positions of PoB kleptoplasts and algal chloroplasts were analyzed using the *rbcL* gene sequences from 114 ulvophycean green algae according to the maximum likelihood (ML) method (*Figure 3B*). A phylogenetic tree was constructed according to the same method used to search for the origins of bacteria-like genes in *H. borneensis* cpDNA.

## Analysis of *H. borneensis* and *B. hypnoides* transcriptomes

The de novo transcript profiles of *H. borneensis* and *B. hypnoides* were obtained from Illumina RNA-Seq data. Total RNA was extracted using the RNeasy Plant Mini Kit (Qiagen). An Illumina library was constructed for each species using the TruSeq RNA Sample Prep Kit. The libraries were sequenced on a HiSeq 2500 platform (101 bp from each end). A total of 290,523,622 reads (29 Gb) and 182,455,350 raw reads (18 Gb) were obtained for *H. borneensis* and *B. hypnoides*, respectively (*Supplementary file 1*). After filtering the low-quality and adapter sequences, the obtained reads were assembled using Trinity version 2.4.0 (*Grabherr et al., 2011*) and clustered using CD-Hit version 4.6 (*Fu et al., 2012*) with the -c 0.95 option. The TransDecoder version 2.0.1 was used to identify 26,652 and 24,127 candidate coding regions from *H. borneensis* and *B. hypnoides*, respectively (*Supplementary file 2*). The gene completeness of the transcripts was estimated using BUSCO version 2.0 (*Waterhouse et al., 2018*). The obtained *H. borneensis* transcripts covered 86.5% (262/303) of the total BUSCO groups, and the *B. hypnoides* transcripts covered 92.7% (281/303) of the conserved genes in Eukaryota (database, eukaryota_odb9). The transcripts were annotated using AHRD version 3.3.3 (https://github.com/groupschoof/AHRD, *Tomato Genome Consortium, 2012*) based on the results of a blastp search against nr, RefSeq, and *Chlamydomonas* proteome dataset on Uni-Prot. The composed functional domains on the transcripts were annotated using InterProScan version 5.23–62 (*Jones et al., 2014*). To distinguish the reliable target species transcripts, the original transcript species were predicted using MEGAN version 5 (*Huson et al., 2007*) and 11,629 and 8630 transcripts were selected as viridiplantal genes. The annotation procedure details are visually presented in *Figure 3—figure supplements 8* and *9*.

A query dataset from the algal transcripts was manually selected to search algae-derived genes on the sea slug DNAs. Then, 176 and 129 transcripts were selected from *H. borneensis* and *B. hypnoides*, respectively. To perform more comprehensive searches, queries from three public genomic datasets derived from *C. lentillifera*, *C. reinhardtii*, and *Cyanidioschyzon merolae* were also obtained; these queries were termed as the A614 dataset (*Supplementary file 3*; *Figure 3—figure supplement 10*).

## Sequencing of the PoB genome

The mean nucDNA size in three PoB individuals was estimated using flow cytometry (*Figure 4—figure supplement 1*). Dissected parapodial tissue (5 mm$^2$) was homogenized in 1 ml phosphate-buffered saline (PBS) containing 0.1% Triton X-100 (Thermo) and 0.1% RNase A (Qiagen) using a BioMasher (Nippi, Tokyo, Japan). The homogenate was then filtered through a 30 μm CellTrics filter (Sysmex, Hyogo, Japan), and the filtrate was diluted with PBS at a density of <5 × 10$^6$ cells/ml. The resulting solution was mixed with genome size standard samples and stained with a 2% propidium iodide solution (SONY, Tokyo, Japan). *Acyrthosiphon pisum* (genome size = 517 Mb) and *Drosophila melanogaster* (165 Mbp) samples were used as genome size standards. The references were processed using the same method described for PoB. The mixture was analyzed on Cell Sorter SH800

(SONY) according to the manufacturer's instructions. The above procedure was repeated for three PoB individuals, and an estimated genome size of 936 Mb was determined.

Genomic DNA was extracted from a single PoB individual using the CTAB-based method (*Murray and Thompson, 1980*). The adapted buffer compositions are summarized in *Supplementary file 16*. A fresh PoB sample (collected on October 17, 2013 and starved for 21 days) was cut into pieces and homogenized in 2× CTAB using a BioMasher. To digest the tissues, 2% vol of proteinase K solution (Qiagen) was added, and the sample was incubated overnight at 55℃. The lysate was emulsified by gentle inversion with an equal volume of chloroform; after centrifugation (12,000 × *g*, 2 min), the aqueous phase was collected using a pipette. This phase was combined with a one-tenth volume of 10% CTAB, mixed well at 60℃ for 1 min, and again emulsified with chloroform. These 10% CTAB and chloroform treatment steps were repeated until a clear aqueous phase was achieved. The aqueous phase was then transferred to a new vessel, an equal volume of CTAB precipitation buffer was overlaid onto the mixture, and the liquids were gently mixed by tapping. The resulting filamentous precipitations (DNA) were removed using a pipette chip and incubated at room temperature for 10 min in high salt TE buffer. DNA was purified according to the protocol supplied with the DNeasy Blood and Tissue Kit (Qiagen); briefly, the supernatant was transferred after vortex mixing, equal volumes of buffer AL (supplied with the kit) and EtOH were added to the supernatant, and the sample was processed on a Qiagen spin column according to the protocol. We finally obtained a DNA quantity of 15 µg from a PoB individual (*Figure 4—figure supplement 11*).

The PoB genomic sequence was obtained via Illumina paired-end and mate-pair DNA sequencing (*Supplementary file 1*; *Figure 4—figure supplement 2*). Two paired-end Illumina libraries containing 250 and 600 bp insertions (DRR029525 and DRR029526) were constructed using a TruSeq DNA Sample Prep Kit (Illumina). Three mate-pair libraries with 3000, 5000, and 10,000 bp insertions (DRR029528, DRR029529, and DRR029530) were constructed using a Nextera Mate Pair Library Prep Kit (Illumina). The libraries were sequenced (150 bp from each end) on a HiSeq 2500 platform (Illumina). A total of 1,130,791,572 and 787,040,878 raw reads were obtained for the paired-end (170 Gb) and mate-pair (118 Gb) libraries. After filtering the low-quality and adapter sequences, the remaining 161 Gb of sequences were assembled using Platanus assembler version 1.2.1 (*Kajitani et al., 2014*) with the default settings. The assembly comprised 8716 scaffolds containing 928,345,517 bp. The repetitive regions were masked using a combination of RepeatModeler version open-1.0.8 and RepeatMasker version open-4.0.5 (http://www.repeatmasker.org). The default parameters were used for identification and masking. A total of 268,300,626 bp (29%) of the assemblies were masked with RepeatMasker.

Strand-specific RNA-Seq sequencing was used for gene modeling (*Supplementary file 1*; *Figure 4—figure supplement 2*). PoB RNA was extracted from an individual after starvation for 20 days (collected on October 17, 2013). TRIzol Plus RNA Purification Kit (Thermo) was used to extract RNA according to the manufacturer's protocol. A paired-end Illumina library (DRR029460) was constructed using the TruSeq Stranded mRNA LT Sample Prep Kit (Illumina). Libraries were sequenced (150 bp from each end) on a HiSeq 2500 platform (Illumina). The library produced a total of 286,819,502 raw reads (28 Gb).

The PoB assemblies were processed using a single transcript-based gene model construction pipeline (AUGUSTUS version 3.2; *Stanke and Morgenstern, 2005*), two transcriptomic data mapping tools (Trinity version 2.4.0 and Exonerate version 2.2.0; *Grabherr et al., 2011*; *Slater and Birney, 2005*), and two non-transcript-based model construction pipelines (GeneMark-ES version 4.33 and glimmerHMM version 3.0.4; *Majoros et al., 2004*; *Ter-Hovhannisyan et al., 2008*). The four obtained gene sets were merged with the EVidenceModeler version 1.1.1 pipeline (*Haas et al., 2008*) to yield a final gene model set. For AUGUSTUS, Braker pipeline version 1.9 (*Hoff et al., 2019*) was used to construct PoB-specific probabilistic models of the gene structure based on strand-specific RNA-Seq data. After filtering the low-quality and adapter sequences, the remaining 181,873,770 RNA-Seq reads (16 Gb) were mapped to the PoB genome assembly using TopHat version 2.1.1 (*Kim et al., 2013*), with the default setting and the Braker pipeline-constructed PoB-specific probabilistic models from mapped read data. TopHat mapped 132,786,439 of the reads (73%) to the PoB model. AUGUSTUS then predicted 78,894 gene models from the TopHat mapping data (as 'splicing junction' data) and the Braker probabilistic model. Using Trinity and Exonerate, de novo transcriptomic data from RNA-Seq data were then constructed, and these were aligned to the genome. Trinity constructed 254,336 transcripts, which were clustered to 194,000 sequences using

CD-Hit version 4.6 (-c 0.95); subsequently, Transdecoder identified 44,596 protein coding regions from these sequences. Exonerate (–bestn 1 –percent 90 options) then aligned 13,141 of the transcripts to the genome. GeneMark-ES with the default settings predicted 107,735 gene models, and glimmerHMM predicted 115,633 models after the model training, with 320 manually constructed gene models from long scaffolds. EVidenceModeler was then used to merge the model with the following weight settings: AUGUSTUS = 9, Exonerate = 10, GeneMark-ES = 1, and glimmerHMM = 2. Finally, EVidenceModeler predicted 77,444 gene models.

The removal of contaminant sequences was minimized to avoid missing horizontally transferred genes. Bacterial scaffolds were defined as those encoding >1 bacterial gene with no lophotrochozoan gene, and the potential bacterial scaffolds were removed from the PoB assemblies. The bacterial genes were predicted using MEGAN software according to a blastp search against the RefSeq database. Of the 40,330 gene hits identified from the RefSeq data, MEGAN assigned the origins for 39,113 genes. Specifically, 719 and 23,559 genes were assigned as bacterial and lophotrochozoan genes, respectively. MEGAN results before removing bacterial scaffolds are summarized in *Supplementary file 17*, with the detailed data of removed scaffolds. Fifty-five of the 8716 scaffolds contained two or more bacterial genes and no lophotrochozoan gene. The mean depth of read coverage of the bacterial scaffolds differed from the mean coverage of sea slug scaffolds.

Kleptoplast- or mitochondria-derived scaffolds were also removed from the assemblies (*Supplementary file 18*). The source of scaffolds was determined based on a blast bit score against the three referential organelle DNAs (kRhip AP014542, kPoro AP014543, and PoB mtDNA AP014544) and the difference of the read depth value from the other (nuclear-derived) scaffolds. The blastn search detected 13 and 1 scaffolds as sequences of kleptoplast or mitochondrial origin, respectively (bit score > 1000; *Supplementary file 18*). Mapping back of the Illumina read (DRR029525) by Bowtie version 2.4.1 indicated that the depth values of the 14 scaffolds were 537 to 5143, and the averaged depth of the other scaffolds was 31 (*Supplementary files 17* and *18*), indicating that the 14 scaffolds were derived from organelle DNAs or repetitive regions on the nucDNA. Mapping of the reads derived from DG (kleptoplast enriched tissue; DRR063263) and parapodium (including a muscle, gonad, and DG; DRR063261) indicated that the relative read depth of the 13 kpDNA-like scaffolds was higher in the DG sample than in the parapodium sample, supporting that the 13 scaffolds are derived from the kleptoplasts (*Figure 4—figure supplement 12*). It was then confirmed that the 14 scaffolds contain no algal nuclear-derived photosynthetic gene using two methods: dot plots with referential organelle DNAs and homology search using A612 query set (*Figure 4—figure supplement 13-15*). Therefore, even if these removed 14 scaffolds are part of the nucDNA, they do not provide HGT evidence. The removed scaffold sequence data were deposited in FigShare (DOI: 10.6084/m9.figshare.12587954).

The final PoB assembly comprised 8647 scaffolds containing 927,888,823 bp (N50 = 1,453,842 bp) and 77,230 genes. Gene completeness was estimated using BUSCO version 2.0 (*Waterhouse et al., 2018*). The predicted gene models were annotated using AHRD version 3.3.3. The results of a blastp search against the SwissProt, Trembl, and *A. californica* proteome datasets on UniProt were used as reference data for AHRD under the following weight parameter settings: SwissProt = 653, Trembl = 904, and *A. californica* = 854. The functional domains were annotated using InterProScan version 5.23–62 (*Figure 4—figure supplement 3*).

A blastp analysis against the RefSeq database was performed to identify algae-derived genes from the constructed genes models. After translating the protein coding region to the amino acid sequence data, the blastp search was adapted to include the '-e-value 0.0001' option. The output was analyzed using MEGAN software with the following LCA and analysis parameters: Min Score = 50, Max Expected = 1.0E-4, Top Percent = 20, Min Support Percent = 0.1, Min Support = 1, LCA percent = 90, and Min Complexity = 0.3.

The GO annotation was assigned using Blast2GO version 5.2.5 according to the blastp searches against the RefSeq database and InterProScan results. SonicParanoid version 1.0.11 was then used for orthogroup detection. All parameters of SonicParanoid were left at the default values. The species analyzed in the orthogroup detections are summarized in *Supplementary file 19*. The phylogenetic tree was constructed using IQ-tree. The resulting trees were visualized using iTol version 4.

Exonerate version 2.2.0 (with the –bestn 1 –model protein2genom options) was used to identify algal genes in the PoB genome. The A614 dataset was used as a query after translating the sequence to amino acids. *C. lentillifera* (green algae) genomic data were used as a control to

estimate the sensitivity of the method. The results were handled and visualized using R (tidyverse packages version 1.2.1).

MMseq2 version 2.6 (–orf-start-mode 1) was used to search for algae-like reads among the trimmed Illumina reads. The matching threshold was set using a default $E$-value < 0.001. As a positive control, PoB genes were selected from the BUSCO analysis of the genomic model data, and 911 gene models detected by BUSCO version two were selected as single-copy orthologs of the metazoa_odb9 gene set and named the dataset P911 (DOI: 10.6084/m9.figshare.13042016).

The HGT index 'h' and the modified index 'hA' were calculated using the R script, HGT_index_-cal.R (https://github.com/maedat/HGT_index_cal copy archived at swh:1:rev:073b9992919b19e0f415aca93c448e073e6b107b; *Maeda, 2021a*; R version 3.6.1). The h-index was calculated as the difference in bit scores between the best prokaryote and best eukaryote matches in the blast alignments, and the hA-index was calculated as the difference in bit scores between the best lophotrochozoan and best algae matches. The blast databases of adapted species are summarized in *Figure 4—source data 1*.

## RNA-Seq analysis of PoB tissues

Total RNA samples from five PoB individuals and one egg mass were obtained for a gene expression analysis. An overview of sample preparation is illustrated in *Figure 4—figure supplement 10*. Collected adult PoB individuals were dissected manually after incubation for 21–94 days. An egg mass was obtained via spontaneous egg lying in an aquarium. A TRIzol Plus RNA Purification Kit (Thermo) was used to extract RNA according to the manufacturer's protocol. Six paired-end and nine single-end Illumina libraries were constructed using a combination of the RiboMinus Eukaryote Kit (Thermo), RiboMinus concentration module (Thermo), and TruSeq RNA Sample Preparation Kit version 2 (Illumina) according to the manufacturers' protocols. Libraries were sequenced (101 bp) on a HiSeq 2500 platform (Illumina). A total of 280,445,422 raw reads (28 Gb) were obtained from the libraries (*Supplementary file 1*).

After filtering the low-quality and adapter sequences, 150,701,605 RNA-Seq reads (13 Gb) were obtained. Only the R1 reads were used for the six paired-end datasets. MMseq2 was used to identify algae-derived reads from the trimmed reads using the A614 dataset as a query. The same parameters were applied as the above-described DNA read search. The PoB gene dataset P911 was also used as a positive control.

The Hisat–stringtie–DESeq2 pipeline was used to conduct a differential gene expression analysis of DGs and DeP (epidermis, muscle, and reproductive systems, DeP). According to the Stringtie protocol manual (http://ccb.jhu.edu/software/stringtie/index.shtml?t=manual), trimmed RNA-Seq reads were mapped to the PoB genome assembly using Hisat version 2.1.0 with the default setting. The obtained BAM files were processed using Stringtie version 1.3.4d (-e option) with PoB gene model data (gff3 format) acquired through the above-mentioned EVidenceModeler analysis. Following the default settings of Stringtie software, if more than 95% of the reads aligned in a gene locus are multi-mapped, processing of that locus is skipped (*Pertea et al., 2015*). It was assumed that this reduces the type I error on the gene enrichment analysis on the duplicated orthogroup. The pairwise sequence similarity of three KRM gene-including orthogroups is summarized on *Supplementary file 21*. The averaged pairwise sequence similarities were 70.3%, 71.0%, and 66.9% in OG0000005, OG0000132, and OG0000446, respectively. The resulting count data were analyzed using R and the DESeq2 package, and 1490 DEGs ($p<0.01$ and $p_{adj}<0.05$) were identified between the tissues. The GOseq (*Young et al., 2010*) and topGO packages in R were used to apply a GO enrichment analysis to the up-regulated genes in DG tissue (threshold: $p<0.01$).

## Sequencing of the *E. marginata* genome

The *E. marginata* genome sequencing process was nearly identical to the methodology applied to PoB. Flow cytometry yielded an estimated genome size of 900 Mb. Genomic DNA was extracted from an individual using a CTAB-based method. Four types of Illumina libraries were constructed: two paired-end libraries with 250 and 500 bp insertions and two mate-pair libraries with 3000 and 5000 bp insertions (*Supplementary file 1*). Using the HiSeq 2500 platform (Illumina), 562,732,268 and 608,977,154 raw reads were obtained for the paired-end (84 Gb) and mate-pair (91 Gb) libraries, respectively. After filtering the low-quality and adapter sequences, the remaining 40 Gb of

sequences were assembled using the Platanus assembler. The assembly comprised 14,285 scaffolds containing 791,005,940 bp.

For gene modeling, strand-specific RNA-Seq sequencing of *E. marginata* was performed. A paired-end Illumina library (DRR029460; see also *Supplementary file 1*) was constructed using a Tru-Seq Stranded mRNA LT Sample Prep Kit (Illumina) and an RNA sample extracted from an *E. marginata* individual via a TRIzol Plus RNA Purification Kit (Thermo). Libraries were sequenced (150 bp from each end) on a HiSeq 2500 platform (Illumina). A total of 286,819,502 raw reads (28 Gb) were obtained from the library. Using the gene modeling procedure described for PoB, EVidenceModeler constructed 71,137 gene models of the genomic assemblies based on the RNA-Seq data.

Next, contaminant-derived bacterial scaffolds were removed from the genomic assemblies. Using the same gene annotation as applied to PoB, it was determined that the 110 of the 14,285 scaffolds contained >1 bacterial gene and no lophotrochozoan gene, and these scaffolds were removed. The organelle-derived scaffolds (kpDNA and mtDNA) were identified using blastn searches and removed from the final assemblies. A blastn search (query = all scaffolds, database = cpDNA of *B. hypnoides* NC_013359.1 or mtDNA of PoB) identified 25 kleptoplast-matching scaffolds and one mitochondria-matching scaffold (bit score > 1000). The complete kpDNA and mtDNA was then reassembled using the same method as described for PoB organellar DNA assembling.

## Ortholog analysis of sacoglossan genes

Orthologous relationships were classified using OrthoFinder version 2.2.3 (*Emms and Kelly, 2015*), and rapidly expanded/contracted families were identified using CAFE version 4.2 (*Han et al., 2013*) based on the OrthoFinder results. All parameters of OrthoFinder were left at the default values, and no 'user-specified rooted species tree' was set. OrthoFinder and CAFE analyses used 16 metazoan species as reference species (*Figure 5—figure supplement 3*). Phylogenetic trees for CAFE were constructed based on 30 single-copy genes in 15 major species according to the OrthoFinder results (*Figure 5—source data 2*). The sequences were trimmed using PREQUAL version 1.02 (*Whelan et al., 2018*), aligned with MAFFT version 7.407, and analyzed with IQ-tree version 1.6.1 for the ML analysis. The obtained ML tree was converted to an ultrametric tree for CAFE analysis based on the divergence times of Amphiesmenoptera-Antliophora (290 Myr) and Euarchontia-Glires (65 Myr) using r8s version 1.81 (*Sanderson, 2003*).

The expanded/contracted families was also analyzed using the *Z*-scores of the assigned gene numbers for each orthogroup. The analysis included all 16 referential species and two sacoglossan species, and an expanded group of 38 was determined on the PoB lineage (Threshold: *Z*-score >2; *Figure 5—figure supplement 5*). The phylogenetic relationships in the orthogroups were analyzed using a PREQUAL-MAFFT-IQ-tree as in the CAFE analysis. The domain structures and gene positions on the constructed genomic data were visualized using the GeneHere script (https://github.com/maedat/GeneHere copy archived at [swh:1:rev:00a122bd16b1cf7efaa60b171047f219e60c9f04; *Maeda, 2021b*]) and Biopython packages.

## Acknowledgements

We appreciate the incisive comments of Masayoshi Kawaguchi, Atsushi J. Nagano, and Kan Tanaka. The sample collection was partially performed by Katsuhiko Tanaka, Tohru Iseto, Rie Nakano, and Hirose Euichi. This work was supported by the MEXT/JSPS KAKENHI (grant numbers 16H06279, 25128713, and 22128001), the NIBB Functional Genomics Facility, the NIBB Data Integration and Analysis Facility, and the Japan Advanced Plant Science Network. Computations were partially performed on the NIG supercomputer at the ROIS National Institute of Genetics.

## Additional information

### Funding

| Funder | Grant reference number | Author |
|---|---|---|
| Japan Society for the Promotion of Science | 25128713 | Taro Maeda |

| Japan Society for the Promotion of Science | 22128001 | Tomoaki Nishiyama Mitsuyasu Hasebe |
| --- | --- | --- |
| Japan Society for the Promotion of Science | 16H06553 | Jun Minagawa |
| Japan Society for the Promotion of Science | 221S0002 | Taro Maeda |
| Japan Society for the Promotion of Science | 17H06388 | Shuji Shigenobu |

The funders had no role in study design, data collection and interpretation, or the decision to submit the work for publication.

## Author contributions

Taro Maeda, Software, Funding acquisition, Visualization, Writing - original draft, Project administration, Design of the study, Photochemical and physiological experiments, Genomic and transcriptomic experiments; Shunichi Takahashi, Resources, Validation, Investigation, Methodology, Writing - review and editing, Photochemical and physiological experiments; Takao Yoshida, Resources, Investigation, Writing - review and editing, Genomics of plastids; Shigeru Shimamura, Software, Investigation, Methodology, Writing - review and editing, Genomics of plastids; Yoshihiro Takaki, Investigation, Methodology, Writing - review and editing, Genomics of plastids; Yukiko Nagai, Investigation, Genomics of plastids; Atsushi Toyoda, Yutaka Suzuki, Resources, Investigation, Genomics of sea slugs; Asuka Arimoto, Resources, Investigation, Algal genomics; Hisaki Ishii, Resources, Investigation, Genomic and transcriptomic experiments; Nori Satoh, Resources, Investigation, Writing - review and editing, Algal genomics; Tomoaki Nishiyama, Methodology, Writing - review and editing, Genomic and transcriptomic experiments; Mitsuyasu Hasebe, Resources, Methodology, Writing - review and editing, Genomic and transcriptomic experiments; Tadashi Maruyama, Conceptualization, Resources, Supervision, Methodology, Writing - review and editing; Jun Minagawa, Conceptualization, Resources, Supervision, Methodology, Writing - review and editing, Photochemical and physiological experiments; Junichi Obokata, Resources, Investigation, Methodology, Writing - review and editing, Genomic and transcriptomic experiments; Shuji Shigenobu, Conceptualization, Resources, Data curation, Supervision, Validation, Investigation, Methodology, Writing - review and editing, Genomic and transcriptomic experiments

## Author ORCIDs

Taro Maeda (iD) https://orcid.org/0000-0003-4185-0135
Atsushi Toyoda (iD) http://orcid.org/0000-0002-0728-7548
Nori Satoh (iD) http://orcid.org/0000-0002-4480-3572
Jun Minagawa (iD) http://orcid.org/0000-0002-3028-3203
Shuji Shigenobu (iD) https://orcid.org/0000-0003-4640-2323

## Decision letter and Author response
Decision letter https://doi.org/10.7554/eLife.60176.sa1
Author response https://doi.org/10.7554/eLife.60176.sa2

# Additional files

## Supplementary files
• Supplementary file 1. Summary of the raw sequencing data.
• Supplementary file 2. Assembly statistics of de novo RNA-seq of donor algae.
• Supplementary file 3. Algal genes used for referential query (A614 dataset).
• Supplementary file 4. Assembly statistics of *Plakobranchus ocellatus* and *Elysia marginata* genome.
• Supplementary file 5. PoB genes having algal top-hit results of blastp. This zip archive contains a file for statistics of PoB genes having algal top-hit results of blastp analysis and a file for detailed annotation of the algal top-hit PoB genes.

• Supplementary file 6. Detailed result of MEGAN analysis.

• Supplementary file 7. Annotation data of the six PoB genes assigned with the child terms of 'Plastid(GO:0009536)'.

• Supplementary file 8. Applied species for HGT index analysis.

• Supplementary file 9. Gene annotation of the DG-up-regulated PoB genes.

• Supplementary file 10. Enriched GOs of the significantly up-regulated genes on the digestive gland of PoB.

• Supplementary file 11. Cross-tabulation table for the significant enrichment analysis of DG-upregulated gene number on the PoB-expanded orthogroup orthologous group.

• Supplementary file 12. Summary of the results of dN/dS-based selection test.

• Supplementary file 13. *E. marginata* scaffolds determined as bacterial contaminants.

• Supplementary file 14. *E. marginata* scaffolds selected as kleptoplast/mitochondrial DNA.

• Supplementary file 15. OrthoFinder result with additional data from *E. chlorotica* gene set (gene counts).

• Supplementary file 16. Composition information of DNA extraction solutions for *P. ocellatus* genomic DN.

• Supplementary file 17. Scaffolds determined as bacterial contaminants during the PoB genome assembling and the result of MEGAN analysis before the removing of the bacterial scaffolds.

• Supplementary file 18. Scaffolds determined as kleptoplast/mitochondrion sequences.

• Supplementary file 19. Analyzed protein sequences by SonicParanoid analysis.

• Supplementary file 20. Result of the Exonerate search. Database, the 13 PoB-derived scaffolds determined as kpDNA; query, algal photosynthetic genes (A614).

• Supplementary file 21. Sequence similarity among the genes belonging OG0000005, OG0000132, and OG0000446.

• Transparent reporting form

### Data availability

All of the raw sequence data obtained in this research have been deposited in the DDBJ Sequence Read Archive (DRA) under BioProject PRJDB4939, PRJDB3267, PRJDB10060, and PRJDB5024. All data collected in this study that are summarized in the figures have been made available on FigShare, at DOI: 10.6084/m9.figshare.12587954 and 10.6084/m9.figshare.13042016. The codes used to analyze HGT index and to visualize gene distribution on scaffolds have been made available on https://github.com/maedat/HGT_index_ca (copy archived at https://archive.softwareheritage.org/swh:1:rev:073b9992919b19e0f415aca93c448e073e6b107b) and https://github.com/maedat/GeneHere (copy archived at https://archive.softwareheritage.org/swh:1:rev:00a122bd16b1cf7efaa60b171047f219e60c9f04).

The following datasets were generated:

| Author(s) | Year | Dataset title | Dataset URL | Database and Identifier |
|---|---|---|---|---|
| Maeda T | 2020 | Genome sequencing of sequestered chloroplast by Plakobranchus ocellatus | https://www.ncbi.nlm.nih.gov/bioproject/PRJDB4939 | NCBI BioProject, PRJDB4939 |
| Maeda T | 2020 | The genome sequencing of Plakobranchus ocellatus (Sacoglossa, Mollusca) | https://www.ncbi.nlm.nih.gov/bioproject/PRJDB3267 | NCBI BioProject, PRJDB3267 |
| Maeda T | 2020 | Tissue specific expression profile of Plakobranchus ocellatus type black | https://www.ncbi.nlm.nih.gov/bioproject/PRJDB10060 | NCBI BioProject, PRJDB10060 |
| Maeda T | 2020 | Transcriptomic profile of Halimeda borneensis (Ulvophyceae, green algae) | https://www.ncbi.nlm.nih.gov/bioproject/PRJDB5024 | NCBI BioProject, PRJDB5024 |

| Maeda T | 2020 | Annotation data set | https://doi.org/10.6084/m9.figshare.13042016 | figshare, 10.6084/m9.figshare.13042016 |
| Maeda T | 2020 | Removed scaffolds from the final P. ocelaltus nuclear genome | https://doi.org/10.6084/m9.figshare.12587954 | figshare, 10.6084/m9.figshare.12587954 |

The following previously published dataset was used:

| Author(s) | Year | Dataset title | Dataset URL | Database and Identifier |
|---|---|---|---|---|
| Arimoto A | 2019 | Caulerpa lentillifera ver. 1.1 | https://marinegenomics.oist.jp/umibudo/viewer/info?project_id=55 | OIST Marine Genomics Unit Genome Sequencing / Annotation Projects, info?project_id=55 |

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
