## [Decision Letter]

**Acceptance summary:**

This work addresses the mechanism underlying kleptoplasty in sea slugs, undoubtedly, a fascinating phenomenon. In the process of kleptoplasty animals transfer food algal chloroplasts into their digestive gland cells and benefit from the continued photosynthesis for up to several months. Previous studies suggested that the incorporation of photosynthesis-related genes from the algal nucleus into the host (sea slug) genomic DNA is a pre-requirement for kleptoplasty to occur. This study shows that such gene transfer is not necessary. The work suggests that also other mechanisms can be in place to establish and maintain the plastid-sea slug relationship. It in turn also implies that the establishment of such connections is likely easier than previously assumed, which should be considered in the context of adaptation mechanisms.

**Decision letter after peer review:**

Thank you for submitting your article "Chloroplast acquisition without the gene transfer in kleptoplastic sea slugs, *Plakobranchus ocellatus*" for consideration by *eLife*. Your article has been reviewed by 3 peer reviewers, and the evaluation has been overseen by a Reviewing Editor and Christian Hardtke as the Senior Editor. The following individuals involved in review of your submission have agreed to reveal their identity: Eugene V Koonin (Reviewer #1); Jia-Xing Yue (Reviewer #3).

The reviewers have discussed the reviews with one another and the Reviewing Editor has drafted this decision to help you prepare a revised submission.

We would like to draw your attention to changes in our revision policy that we have made in response to COVID-19 (https://elifesciences.org/articles/57162). Specifically, we are asking editors to accept without delay manuscripts, like yours, that they judge can stand as *eLife* papers without additional experimental data, even if they feel that they would make the manuscript stronger. However, there are several issues raised by the reviewers that we ask you to clarify, before we can take a final decision on your work.

Summary:

The manuscript addresses interesting evolutionary questions and hints at possible mechanisms about the phenomenon that certain sea slugs can take advantage of algal photosynthesis by maintaining their chloroplasts long after having ingested algae. The authors present high-quality genome assemblies and address the question of whether algal to slug HGT may have facilitated sequestration of algal chloroplasts and find no evidence to support this. There appear to be other genomic changes in the slug genome that could support retention of the chloroplasts.

Essential revisions:

1. The candidate kleptoplasty-related molluscan (KRM) genes that the authors identified in this study serve as an important stepping stone for direct assessment of their kleptoplasty-related function in future. That being said, their functional involvement is far from being proven. The authors demonstrate significant up-regulation of the expression of a set of nearly 200 genes in kleptoplast-harboring slugs. There are some functional trends among these genes, in particular, the enrichment of proteases and carbohydrate metabolism proteins, but it is unclear how this could possibly solve the enigma of the long-term persistence of the functioning kleptoplast given the short life time of the photosynthetic proteins. It is fine to present the data on gene expression up-regulation and the complementary evolutionary analysis, but the authors should be far more circumspect in their conclusions on the direct relevance of the respective genes and rather admit that the puzzle remains.

2. We think that naming the phenomenon of the slugs retaining chloroplasts "DNA/RNA-independent transformation" is problematic, because of the fixed definition of the word "transformation" in (molecular) biology, where transformation is defined as a genetic alteration, which it is not. It does not appear to be plausible that the phenotype and organelle "have moved beyond the species" – they are not inherited in the new species and it is a dead-end for these chloroplasts that are eaten and die a few months later than the rest of the algae. There is likely selective pressure for the slug to hold on to its "food" for a while longer due to the chemical activity of the chloroplast, but there is no selective pressure on the chloroplast to be consumed in this way that would constitute reciprocal evolutionary changes. We do not see this as anything beyond animal adaptations that facilitate utilization of a food source.

3. "Taken together, the data for the three photosynthetic indexes indicate that kleptoplast photosynthesis increases resistance to starvation in PoB." – This isn't directly shown here. There are no plastid-free control slugs. Light exposed slugs live longer – but how do we know it is photosynthesis? Light can affect other cellular processes – ROS, etc.

4. In the photosynthesis experiments (Figure 2), the authors compare the kleptoplast activity to the plastid in the algae H. borneensis. However, genome sequencing revealed two distinct plastid genomes (kRhip and kPoro) which corresponded with plastids from Rhipidosiphon lewmanomontiae and Poropsis spp., neither of which were included in genomic comparisons, and were not used in the comparison for photosynthetic rate.

5. Have the authors considered that the slugs may have acquired photosynthesis related genes from non-algal origins that could potentially support the kleptoplasts? There are other examples of symbioses with many partners where HGTs not from any of the original lineages support the relationship (Husnik, 2013, Cell). Line 170 mentions 6 Chloroplast-related genes in PoB but there is no follow up on what these are, only that they do not look like algae. Could these support the photosynthetic abilities of the kleoptoplasts?

6. In general, the manuscript's readability needs improvement. This is particularly severe for the referral to the figures and a lack of clarity for the RNA seq analyses.

- Different panels of main figures (e.g. Figure 1A-1F, Figure 2B) are not well-referred in the current manuscript, which compromised the readability of the current manuscript.

- Figure 5C is too complex and busy. We recommend to simplify this panel. Here are some suggestions regarding its current form: The authors could consider further subdividing it into multiple panels. Also, the "domain structure" key boxes should belong to the left side of this panel , whereas an additional key box should be provided for explaining colors used in the gene position graph (on the right side of this panel). Otherwise, the readers could easily get confused. Finally, the authors might want to consider changing the color scheme used for FDR to make it more different from the color schemes used for denoting domain structure and gene position.

- Some of the RNASeq analyses are difficult to follow. For instance trying to match Figure S22 with the text is difficult. The methods states that 5 slugs were used for whole-transcriptome sequencing, but 15 libraries were constructed, and there are only 13 library IDs in Figure S22. There are 15 libraries in the heat maps in the main text. What model was used in DESeq2 to call differential expression? What level of coverage was achieved in the RNA-Seq libraries? It seems there is no biological replication for some of the samples?

7. It is interesting that the genes that are expanded in copy number also look to be upregulated. However, it is unclear whether or not multiple mappings are accounted for, and how. If multiple mappings of closely related genes are not properly accounted for then it may appear as up-regulation. Similarly, how close in sequence are the paralougs of these genes, ie, would the reads map to each other?

8. During genome assembly, were the removed "bacterial scaffolds" also present at different coverage levels? Did the authors screen for fungal contaminants also? They mention in the RNA-Seq analyses how other food/biofilms can be present, so there it is likely that other contaminants could be in the assembly.

9. Figure 3C: According to the authors, "the vertical bar chart indicates the number of genes conserved among the species", but if so, why do rare genes seem to be more conserved than core genes according to this bar chart? This needs to be clarified.

10. To test if duplicated genes in PoB are under diversifying selection (a.k.a positive selection) or relaxed purifying selection during neofunctionalization, the authors could use tools such as PAML for a formal test.

---

## [Author Response]

Essential revisions:1. The candidate kleptoplasty-related molluscan (KRM) genes that the authors identified in this study serve as an important stepping stone for direct assessment of their kleptoplasty-related function in future. That being said, their functional involvement is far from being proven. The authors demonstrate significant up-regulation of the expression of a set of nearly 200 genes in kleptoplast-harboring slugs. There are some functional trends among these genes, in particular, the enrichment of proteases and carbohydrate metabolism proteins, but it is unclear how this could possibly solve the enigma of the long-term persistence of the functioning kleptoplast given the short life time of the photosynthetic proteins. It is fine to present the data on gene expression up-regulation and the complementary evolutionary analysis, but the authors should be far more circumspect in their conclusions on the direct relevance of the respective genes and rather admit that the puzzle remains.

We appreciate the editor's comment on this point. In accordance with the editor's comment, we emphasized that the direct relevance of the potential KRM genes for kleptoplasty remains unclear.

Revised version:

“The comparative genomic and transcriptomic analyses of these species demonstrate the complete lack of photosynthetic genes in these sea slug genomes and supported an alternative hypothetical kleptoplasty mechanism.”

Revised version:

“The combination of genomics and transcriptomics suggests that the maintenance of algae-derived protein activity is the most probable mechanism for retaining PoB photosynthesis. Because of the limited longevity of the photosynthetic proteins in algal cells and/or in vitro (Roberts et al., 2013), previous studies have discussed the elongation of algal protein lifespans via protective sea slug proteins as an alternative hypothesis to HGT (de Vries and Archibald, 2018; Serôdio et al., 2014). This study shows three types of potential KRM genes: apolipoprotein D-like, cathepsin D-like, and lectin-like genes. Some of these KRM genes may encode proteins that protect algae-derived proteins. Although further studies are needed to elucidate their functional roles in detail, the obtained results are consistent with HGT’s alternative hypothesis, algal protein retention.

Previous RNA-Seq studies of *Elysia timida* and *E. chlorotica* found up-regulation of superoxide dismutase (SOD) genes in response to photostress and postulated that SOD protects algal proteins in the kleptoplasts from oxidative damage (Chan et al., 2018; de Vries et al., 2015). No significant up-regulation of the SOD gene in PoB DG was found (Figure 5—figure supplement 1). However, there was up-regulation of apolipoprotein D-like genes and the expansion of these genes in PoB lineages (Figure 5). Apolipoprotein D, a lipid antioxidant, confers resistance to oxidative stress in higher plants and animal brains (Bishop et al., 2010; Charron et al., 2008). Ortholog analysis of the new *E. chlorotica* data found no gene number expansion of these homologs in *E. chlorotica*, i.e., only three apolipoprotein D-like homologs (Supplementary File 15). Although the retention process details may differ among species and abiotic conditions, it is attractive to speculate that oxidative stress resilience is of major importance for kleptoplasty in multiple sacoglossan species.

Previous kleptoplasty studies have not focused on cathepsin D-like and lectin-like genes. The involvement of the mechanisms of these proteins in kleptoplasty remains to be solved. Cathepsin D degrades intracellular proteins and contributes to the degradation of damaged mitochondria (Benes et al., 2008). In general, damaged photosynthetic proteins generate abundant ROS, promoting further protein damage. In PoB, cathepsin D-like protease may degrade damaged proteins and prevent further protein inactivation. Lectin is involved in self‐recognition in natural immunity (Geijtenbeek and Gringhuis, 2009; Worthley et al., 2005). The diverse lectins expressed in PoB DG tissue may bind the antigens of algae-derived molecules, mediate the detection of non-self-proteins and/or saccharides, and lead to the selective degradation and retention of algae-derived proteins and organelles.”

2. We think that naming the phenomenon of the slugs retaining chloroplasts "DNA/RNA-independent transformation" is problematic, because of the fixed definition of the word "transformation" in (molecular) biology, where transformation is defined as a genetic alteration, which it is not. It does not appear to be plausible that the phenotype and organelle "have moved beyond the species" – they are not inherited in the new species and it is a dead-end for these chloroplasts that are eaten and die a few months later than the rest of the algae. There is likely selective pressure for the slug to hold on to its "food" for a while longer due to the chemical activity of the chloroplast, but there is no selective pressure on the chloroplast to be consumed in this way that would constitute reciprocal evolutionary changes. We do not see this as anything beyond animal adaptations that facilitate utilization of a food source.

Given the reviewer's comment, we have replaced the term "DNA/RNA-independent transformation" throughout the paper with "phenotype acquisition" to use more precise terms. We have also emphasized that the organellar acquisition is a temporary phenomenon as below.

Revised version:

“However, these two examples of DNA/RNA-independent phenotype acquisition are limited to the transfer of simple phenotypes that depend on just a few enzymes. In contrast, the well-known complexity of photosynthesis suggests that sea slug kleptoplasty depends on DNA/RNA-independent phenotype acquisition of complex pathways requiring multiple enzymes (e.g., entire photosystems and the Calvin cycle).”

Revised version:

“However, these studies indicate that PoB gains adaptive photosynthetic activity without acquiring any of the many algal nucleic genes involved in photosynthesis. This is evidence of a phenotype (and organelle) acquisition, although temporally, from the donor without DNA or RNA transfer.”

3. "Taken together, the data for the three photosynthetic indexes indicate that kleptoplast photosynthesis increases resistance to starvation in PoB." – This isn't directly shown here. There are no plastid-free control slugs. Light exposed slugs live longer – but how do we know it is photosynthesis? Light can affect other cellular processes – ROS, etc.

We thank the reviewer for this comment. Given the reviewer's comment, we have changed the text about the relationships between the kleptoplast photosynthesis and starvation resistance. We have included a new citation about another sacoglossan species having a shorter photosynthetic maintenance period. We agree that additional information on plastid-free control slugs, as the reviewer suggested, would be valuable. Regrettably, however, we would be unable to do the experimentation, because it is technically impossible to obtain kleptoplast-free healthy individuals due to the long-term kleptoplast retention period of PoB. As reviewers suggest, the production of reactive oxygen species (ROS) is a typical biochemical reaction against light illumination. However, we consider that ROS, which is toxic to many organisms, is an inadequate explanation for the presented results. ROS production is expected to shorten the survival period of illuminated PoBs. In our understanding, the simplest positive effect of light on growth is nutrient production through photosynthesis. Although several organisms have light-dependent secondary metabolites (e.g., vitamin D synthesis in vertebrates), the importance of such metabolites has rarely been studied in sea slug, and further research is needed to discuss the impact of their production.

Revised version:

“Although a study using *P.* cf. *ocellatus* reported that photosynthesis had no positive effect on the survival rate (Christa et al., 2014c), our results indicated that this finding does not apply to PoB.

Given the three photosynthetic indices’ data, we concluded that the increase in PoB survival days was due to photosynthesis. A previous study using the short-term (retention period of 4–8 days) kleptoplastic sea slug, *Elysia atroviridis*, indicated no positive effect of light illumination on their survival rate (Akimoto et al., 2014). These results supported that light exposure does not affect sacoglossan longevity in the absence of kleptoplasts. Although plastid-free PoB may directly indicate that kleptoplasty extends longevity, there is no way to remove kleptoplasts from PoB, except during long-term starvation. PoB feeds on nothing but algae and retains the kleptoplast for 3 to 5 months. Although future research methods may allow for more experimental analyses of kleptoplast functions, the currently most straightforward idea is that kleptoplast photosynthesis increases the starvation resistance of PoB.”

4. In the photosynthesis experiments (Figure 2), the authors compare the kleptoplast activity to the plastid in the algae H. borneensis. However, genome sequencing revealed two distinct plastid genomes (kRhip and kPoro) which corresponded with plastids from Rhipidosiphon lewmanomontiae and Poropsis spp., neither of which were included in genomic comparisons, and were not used in the comparison for photosynthetic rate.

We agree that this point requires clarification. We have added the text to the Results and the Materials and methods. As mentioned by the editors, the algal species used in the photosynthetic analysis (*H. borneensis*) do not match the species detected in the kleptoplast genome sequencing (*R. lewmanomontiae* and *Poropsis* spp.). This discrepancy is due to the thallus of *R. lewmanomontiae* and *Poropsis* spp were too small for our photosynthesis experiments. Instead of the two algal species, we selected *H. borneensis* from the already known donors of PoB kleptoplasts because its thallus is large enough for photosynthesis analysis.

Revised version:

“These values were only slightly lower than those of healthy *Halimeda borneensis*, a kleptoplast donor of PoB (Maeda et al., 2012), which showed Fv/Fm values of 0.73 to 0.76. The donor algae of PoB consisted of at least eight green algal species, and they are closely related to *H. borneensis* (Maeda et al., 2012). Although it is not clear whether the Fv/Fm was the same for all donor algae, in healthy terrestrial plants, Fv/Fm is almost identical (~0.83) regardless of species (Maxwell and Johnson. 2000), and the values of *H. borneensis* were similar to those of other green algae (e.g., *Chlamydomonas reinhardtii*, Fv/Fm = 0.66–0.75; Bonente et al. 2012). Hence, we assumed that Fv/Fm values are similar among the donor species. The Fv/Fm value suggested that PoB kleptoplasts retain a similar photochemical efficiency of PSII to that of the food algae for more than 3 months.”

Revised version:

“The seawater was not changed during the incubation period, except to adjust the salinity using distilled water. Photosynthetic activity indices (oxygen generation rate and PAM fluorometry) were measured using oxygen sensor spots (Witrox 4; Loligo Systems, Tjele, Denmark) and PAM-2500 (Walz, Effeltrich, Germany), respectively (Figure 2—figure supplement 1B–F). *H. borneensis*, which was easily collected and kept in the laboratory, was used as a reference for comparison in the PAM analysis. This species is a donor alga of PoB kleptoplasts and closely related species with the major donors of PoB, *R. lewmanomontiae* and *Poropsis* spp. These two major donors firmly adhered to the rock, were tangled up in other algal species, and were difficult to be collected.”

5. Have the authors considered that the slugs may have acquired photosynthesis related genes from non-algal origins that could potentially support the kleptoplasts? There are other examples of symbioses with many partners where HGTs not from any of the original lineages support the relationship (Husnik, 2013, Cell). Line 170 mentions 6 Chloroplast-related genes in PoB but there is no follow up on what these are, only that they do not look like algae. Could these support the photosynthetic abilities of the kleoptoplasts?

We appreciate the reviewer's comment on this point. In our previous MS, we had conducted a MEGAN analysis and confirmed that there were no genes derived from algae or plants in the PoB gene model. Because we had included all photosynthetic organisms (except land plants) as "algae", we had considered these results indirectly indicated that there were no genes related to photosynthesis. However, as pointed by the editors, we did not mention the presence or absence of photosynthesis-related genes based on their predicted functions. Therefore, in this revised MS, we have added a detailed section about the potential HGT from non-algal species (MEGAN analysis) and the functional feature of the gene having non-algal origin (Supplementary file 5 and Supplementary file 6). We also added descriptions and a table (Supplementary file 7) about six Chloroplast-related genes. As a consequence, we were not able to find any genes functionally related to photosynthesis via the functional analysis.

Revised version:

“A blastp search using the A614 dataset, which contains sequences of the potential gene donor (e.g., transcriptomic data of *H. borneensis*), also determined no positive HGT evidence (Supplementary file 5).

Using LCA analysis for all PoB gene models, some genes were predicted to originate from species other than Lophotrochozoa. These genes may be due to HGT, but no photosynthesis-related genes were found from them. MEGAN predicted that 20,189 of the PoB genes originated from Lophotrochozoa and its subtaxa. The prediction also assigned 5550 genes to higher adequate taxa as Lophotrochozoa (e.g., Bilateria). MEGAN assigned the 2964 genes to Opisthokonta other than Lophotrochozoa (e.g., Ecdysozoa) and 312 genes derived from proteobacteria as promising horizontally transferred genes. However, these 3276 genes contained no photosynthetic gene. Many of the PoB genes assigned as promising horizontally transferred genes encoded reverse transcriptase. For the remaining 48,215 genes, MEGAN assigned no species. A homology search against the public database annotated 41,203 of the no-taxon-assigned genes as functionally unknown genes (FigShare; DOI: 10.6084/m9.figshare.13042016). Other no-taxon-assigned 7012 genes were not associated with photosynthesis, except one gene (p288c60.92) annotated to “photosystem I assembly protein ycf3.” However, this annotation to p288c60.92 was not reliable because it seems to be derived from incorrect annotations on public databases. Our reannotation via blastp search against the RefSeq database indicated the similarity of the PoB gene (p288c60.92) to “XP_013785360, death-associated protein kinase related-like” of a horseshoe crab. The details of the analyses are summarized in Supplementary File 6. MEGAN-based analysis, hence, indicated that several non-photosynthetic genes might have originated from proteobacteria or other organisms but provided no evidence of algae-derived HGT.

In gene function assignment with Gene Ontology (GO) terms, no PoB gene model was annotated as a gene relating “Photosynthesis (GO:0015979),” although the same method found 72 to 253 photosynthesis-related genes in the five referential algal species (Figure 4A). Six PoB genes assigned to the child terms of “Plastid (GO:0009536)” were found (Figure 4A; Supplementary File 7). However, an ortholog search with animal and algal genes did not support these six genes’ algal origin (Figure 4—figure supplement 5). It was considered that the sequence conservation beyond the kingdom caused these pseudo-positives in the GO assignment. The function of these six genes had no relationships to photosynthesis (Supplementary File 7); six genes relate proteasome (p2972c65.3), arginine kinase (p197c68.18), DNA-binding response regulator (p234c64.89), chromatin structure (p466c59.83), mitochondrial inner membrane translocase (p503c65.126), and functionally unknown protein (p45387c41.1).”

6. In general, the manuscript's readability needs improvement. This is particularly severe for the referral to the figures and a lack of clarity for the RNA seq analyses.

We changed the wording and other details of the manuscript to make it as readable as possible.

- Different panels of main figures (e.g. Figure 1A-1F, Figure 2B) are not well-referred in the current manuscript, which compromised the readability of the current manuscript.

We clarified which sub-panels of the figure correspond to the text and added descriptions on the legends as below. We also changed panels' ordering in Figure 1 accordingly.

Revised version:

“Chloroplast sequestration in sea slugs has attracted much attention due to the uniqueness of the algae-derived phenotype acquisition. Some species of sacoglossan sea slugs (Mollusca: Gastropoda: Heterobranchia) can photosynthesize using the chloroplasts of their algal food (Figure 1A and B Figure ; de Vries and Archibald, 2018; Kawaguti, 1965; Pierce and Curtis, 2012; Rumpho et al., 2011; Serôdio et al., 2014).

Revised version:

“The retention period of photosynthesis differs among sacoglossan species (1 to >300 days; Figure 1B; Christa et al., 2015, 2014a, 2014b; Evertsen et al., 2007; Laetz and Wägele, 2017) and development stages and depends on the plastid “donor” species (Curtis et al., 2007; Laetz and Wägele, 2017).”

Revised version:

“Here, the genome sequences of another sacoglossan species, *Plakobranchus ocellatus* (Figure 1C–E) , are presented to clarify whether HGT is the primary system underlying kleptoplasty.”

Revised version:

“However, a recent phylogenetic analysis showed that *P. ocellatus* is a species complex (a set of closely related species; Figure 1F;Figure Christa et al., 2014c; Krug et al., 2013; Maeda et al., 2012; Meyers-Muñoz et al., 2016; Yamamoto et al., 2013).”

Revised version:

“Based on the measurement of oxygen concentrations in seawater, starved PoB individuals (“d38” and “d109”) displayed gross photosynthetic oxygen production (Figure 2B and C).”

Revised version:

“Figure 1. Kleptoplasty in sea slugs.

(A) Process of algal chloroplast retention by a sacoglossan sea slug (Pierce and Curtis, 2012). Sacoglossan sea slugs puncture the cell wall of food algae to suck out the protoplasm. The chloroplasts in the protoplasm are transported to the sea slug’s intestinal tract, and the intestinal epithelial cells sequester chloroplasts by phagocytosis. The sequestered chloroplasts (kleptoplasts) maintain the photosynthetic activity in the cell for days to months. The sacoglossan cell contains no algal nuclei. Kleptoplast has never been found in germ cells of sea slug. (B) Phylogenetic distribution of kleptoplasty in the order Sacoglossa. Phylogenetic analysis showed that a common ancestor of Sacoglossa acquired non-functional chloroplast retention phenomena (without the maintenance of photosynthetic function), and multiple sacoglossan groups subsequently acquired the ability to maintain photosynthetic activity. Phylogenetic tree and kleptoplasty states are simplified from Christa et al. (2015). Christa et al. (2014b) defined functional chloroplast retention for less than two weeks as "short term retention", and for more than 20 days as "long term retention". Relationships within Heterobranchia are described according to Zapata et al. (2014). The red-colored taxa include the species used in the present study (*P. ocellatus* and *E. marginata*). (C–E) Photo images of PoB starved for 21 days. (C) Dorsal view. H, head; R, rhinophores; P, parapodia (lateral fleshy flat protrusions). Almost always, PoB folded parapodia to the back in nature. (D) The same individual of which parapodium was turned inside out (without dissection). The back of the sea slug and inside of the parapodia are green. This coloration is caused by the kleptoplasts in DG, which are visible through the epidermis. (E) Magnified view of the inner surface of parapodium. The diagonal green streaks are ridged projections on the inner surface of the parapodium. The cells containing kleptoplasts are visible as green spots. (F) Phylogeny of the *P.* cf. *ocellatus* species complex based on mitochondrial *cox1* genes (ML tree from 568 nucleotide positions) from INSD and the whole mtDNA sequence. The sequence data for the phylogenetic analysis are listed in Figure 1–source data 1. Clade names in square brackets are based on Krug et al. (2013). Asterisks mark the genotypes from Krug et al. (2013). Study topics analyzed by previous researchers were described within the colored boxes for each cluster. Small black circles indicate nodes supported by a high bootstrap value (i.e., 80–100%). *Thuridilla gracilis* is an outgroup. *Plakobranchus papua* is a recently described species and previously identified as *P. ocellatus* (Meyers-Muñoz et al., 2016).”

- Figure 5C is too complex and busy. We recomment to simplify this panel. Here are some suggestions regarding its current form: The authors could consider further subdividing it into multiple panels. Also, the "domain structure" key boxes should belong to the left side of this panel , whereas an additional key box should be provided for explaining colors used in the gene position graph (on the right side of this panel). Otherwise, the readers could easily get confused. Finally, the authors might want to consider changing the color scheme used for FDR to make it more different from the color schemes used for denoting domain structure and gene position.

Given the reviewer's comment, we have split up Figure 5 into 2 figures; Figure 5 (containing Figure 5A and B) and Figure 6 (renamed from Figure 5C). In Figure 6 (Figure 5C), we also changed the color schemes used for gene position, following the suggestion. Moreover, we removed a part illustrating the domain structure for simplicity. The domain structure information can be seen on Supplementary Figures in Figure 6 (Figure 6—figure supplement 1 and Figure 6—figure supplement 3). On the Supplementary Figures, we changed the key box positions and added descriptions explaining colors' meaning.

Revised version:

“Figure 6. Evolutional process of KRM gene candidates of OG0000132

(A) ML phylogenetic tree for OG0000132 genes (for details, see Figure 6—figure supplement 1). Red circle, PoB gene; blue square, *E. marginata* (Ema) gene. Genes mentioned in the text are indicated with their IDs beside the circles. Pictograms represent non-sacoglossan species defined in Figure 5—source data 2. (B) Expression of each PoB gene and gene position on the genomic assemblies. The left-side heat plot indicates normalized expression degree (FPKM) of the PoB genes in various tissues; tissue abbreviations are defined in Figure 4 and Figure 4—figure supplement 10. The vertical gene order corresponds with the position in Figure 6A. The white box with a diagonal line means a non-PoB gene (no expression data is available). Tissue samples for the FDR calculation (DG versus DeP) were enclosed with square brackets. Color scales for FPKM and FDR are visualized on the right-side boxes. The FDR’s black color indicates that the value could not be calculated (NA) due to its undetectable gene expression. The right panel shows the genomic positions of the genes. The bands’ color indicates the correlation between the gene and position, purple text and arrows indicate the scaffold ID and direction, and colored boxes on the scaffolds mean the genes’ positions (magenta, OG0000132 gene; gray box, other protein-encoding genes). Scaffolds having less than five OG0000132 genes were omitted from the figure.” Figure

- Some of the RNASeq analyses are difficult to follow. For instance trying to match Figure S22 with the text is difficult. The methods states that 5 slugs were used for whole-transcriptome sequencing, but 15 libraries were constructed, and there are only 13 library IDs in Figure S22. There are 15 libraries in the heat maps in the main text. What model was used in DESeq2 to call differential expression? What level of coverage was achieved in the RNA-Seq libraries? It seems there is no biological replication for some of the samples?

We appreciate the reviewer's comment on this point. The reviewer's comment is correct. In the previous version, we missed descriptions for the two libraries in Figure S22. We have added a description of these two libraries, the model used for DEseq2, level of coverage, and biological replication statement to Figure 4—figure supplement 10 (renamed from Figure S22).

Revised version:

“Figure 4—figure supplement 10

Overview of the sample preparation for *P. ocellatus* type black transcriptomic analysis

(A–C) Photographs of the PoB dissection and the positions of sample tissues. (A) Dorsal views of a fresh PoB individual with spread (or inside-out) parapodia. (B) The upper plate shows an enlarged view of the tissue dissected from the parapodium (-PA sample). The plate on the lower left shows the parapodium, excluding the digestive gland part, collected in a plastic tube (-DeP sample). The lower-right plate shows the digestive gland cut out from the parapodium using a razor (-DG sample). (C) Egg mass of PoB. (D) Incubation states of the dissected sea slugs and the relationships among the samples, libraries, and RNA-seq analysis. "Mean coverageData" means the averaged depth of coverage of each library to the PoB genome. The libraries from three individuals (T1–3) were used for DEG calling using the Wald test.”

7. It is interesting that the genes that are expanded in copy number also look to be upregulated. However, it is unclear whether or not multiple mappings are accounted for, and how. If multiple mappings of closely related genes are not properly accounted for then it may appear as up-regulation. Similarly, how close in sequence are the paralougs of these genes, ie, would the reads map to each other?

The editors have raised an important point; however, we believe that our analysis adequately considers the effect of multiple mapping. Adopted software excludes multiple mapping regions from the analysis, and there are large sequence differences among the PoB expanded genes.

We have used "Stringtie" software to convert Hisat2 bam data to mapping count data. This software skips the analysis of multi-mapped locus. In their original paper about Stringtie, authors described, "Note that if more than a certain percentage of the reads (by default 95%) aligned in a gene locus are multi-mapped, then StringTie will skip processing that locus." (Pertea et al., 2015 *Nature Biotechnology*). Hence, the highly resemble genes were removed from the analysis via Stringtie. We assumed that this handling will reduce the type I error on our gene enrichment analysis on the duplicated orthogroup.

We have also described the condition of the sequence similarity among the paralogs as belows. The similarity (pairwise percent similarly) in the KRM-having orthogroup (OG0000005, OG0000132, and OG0000446) was summrized on Supplementary file 21. The averaged pairwise sequence similarities were 70.3%, 71.0%, and 66.9% in OG0000005, OG0000132, and OG0000446, respectively. These differences suggest that multiple mapping is rare among the paralgos.

Revised version:

“The obtained BAM files were processed using Stringtie version 1.3.4d (-e option) with PoB gene model data (gff3 format) acquired through the above-mentioned EVidenceModeler analysis. Following the default settings of Stringtie software, if more than 95% of the reads aligned in a gene locus are multi-mapped, processing of that locus is skipped (Pertea et al. 2015). It was assumed that this reduces the type I error on the gene enrichment analysis on the duplicated orthogroup. The pairwise sequence similarity of three KRM gene-including orthogroups is summarized on Supplementary File 21. The averaged pairwise sequence similarities were 70.3%, 71.0%, and 66.9% in OG0000005, OG0000132, and OG0000446, respectively. The resulting count data were analyzed using R and the DESeq2 package, and 1490 DEGs (*p* < 0.01 and *p*_adj_ < 0.05) were identified between the tissues. The GOseq (Young et al., 2010) and topGO packages in R were used to apply a GO enrichment analysis to the up-regulated genes in DG tissue (threshold: *p* < 0.01).”

8. During genome assembly, were the removed "bacterial scaffolds" also present at different coverage levels? Did the authors screen for fungal contaminants also? They mention in the RNA-Seq analyses how other food/biofilms can be present, so there it is likely that other contaminants could be in the assembly.

We agree that this point requires clarification. We have added sequence file of the removed bacterila scaffolds and a description of summarized coverage levels for bacterial assembly in the table (Supplementary file 17). We found no scaffold indicating the similar mean coverage of reliable sea slug scaffolds (31, SD = 7). The mean coverages of the bacterial scaffolds were around 10 (3-23), althogh a scaffold (scaffold4993_cov50) indicated higher coverage than sea slug scaffolds (328). About fungal contamination, because too much sensitive removal may exclude HGT regions as contaminant sequences, we minimized the removal of potential bacterial contamination and skipped the removal of potential fungal contamination.

Revised version:

“GeneMark-ES with the default settings predicted 107,735 gene models, and glimmerHMM predicted 115,633 models after the model training, with 320 manually constructed gene models from long scaffolds. EVidenceModeler was then used to merge the model with the following weight settings: AUGUSTUS = 9, Exonerate = 10, GeneMark-ES = 1, and glimmerHMM = 2. Finally, EVidenceModeler predicted 77,444 gene models.

The removal of contaminant sequences was minimized to avoid missing horizontally transferred genes. Bacterial scaffolds were defined as those encoding >1 bacterial gene with no lophotrochozoan gene, and the potential bacterial scaffolds were removed from the PoB assemblies. The bacterial genes were predicted using MEGAN software according to a blastp search against the RefSeq database. Of the 40,330 gene hits identified from the RefSeq data, MEGAN assigned the origins for 39,113 genes. Specifically, 719 and 23,559 genes were assigned as bacterial and lophotrochozoan genes, respectively. MEGAN results before removing bacterial scaffolds are summarized in Supplementary file 17 with the detailed data of removed scaffolds. Fifty-five of the 8716 scaffolds contained two or more bacterial genes and no lophotrochozoan gene. The mean depth of read coverage of the bacterial scaffolds differed from the mean coverage of sea slug scaffolds.”

9. Figure 3C: According to the authors, "the vertical bar chart indicates the number of genes conserved among the species", but if so, why do rare genes seem to be more conserved than core genes according to this bar chart? This needs to be clarified.

We are concerned that the editors may have misunderstood "rare genes". In the previous manuscript, we defined the "rare genes" as below, "Rare gene, determined from a single or no Bryopsidales species". The second vertical bar from the left shows the number of commonly retained genes in Cyme (*C. merolae*) and Vali (*V. litorea*). Cyme and Vali are not Bryopsidales species (Rhodophyceae and Stramenopiles, respectively). The cpDNAs of Cyme and Vali contain many genes that are not retained in Bryopsidales plastids, which results in a higher vertical bar. However, these genes are not present in Bryopsidales (see intersect connectors), so we have assigned them as "rare genes" in our classification. We have added a description on these points to the legend.

Revised version:

“Gray shading indicates non-Viridiplantae algae, and magenta shading indicates PoB kleptoplasts. Cyme (*C. merolae*) and Vali (*V. litorea*) had more than 100 genes that Bryopsidales does not have (e.g., left two vertical bars). (D) Box plots of tblastn results.”

10. To test if duplicated genes in PoB are under diversifying selection (a.k.a positive selection) or relaxed purifying selection during neofunctionalization, the authors could use tools such as PAML for a formal test.

The results of the additional analysis (PAML and RELAX) showed that several branches of the expanded genes have evolved under relaxed selection relative to the rest branches. We have added the following text to the manuscripts. The obtained PAML and RELAX results were summarized in Supplementary file 12.

Revised version:

“Therefore, we considered that (1) the ubiquitously expressed p609c69.52 gene in α-clade is a functional ortholog of the mammalian cathepsin D gene, (2) the p374c67.53 gene in β-clade relates sea slug embryo development, and (3) the γ-clade genes have been acquired with the development of plastid sequestration. To test the positive or relaxed selection on the duplication event, the ratio of substitution rates at non-synonymous (dN) and synonymous (dS) sites (ω = dN/dS) was used. Longly alignable heterobranchian genes from OG0000132 were used and statistically tested using CodeML and RELAX software (Supplementary file 12). The CodeML analysis (branch-site model) determined no significant positive selection on the basal node of the γ-clade (*p* > 0.05, *χ*^2^ test), but RELAX suggested that the γ-clade evolved under relaxed selection relative to the rest branches (*K* = 0.44, *p* < 0.001). The cathepsin D-like genes formed multiple tandem repeat structures in the PoB genome, although other Heterobranchia had no tandem repeat (Figure 6B; Figure 6—figure supplement 1 and Figure 6—figure supplement 2).”

Revised version:

“The gene duplication in OG0000446 seems to have happened in the PoB lineage and at the node between PoB and *E. marginata* (Figure 5B; Figure 6—figure supplement 3 and Figure 6—figure supplement 4). The sacoglossan genes were duplicated in a monophyletic clade (clade I) only, and all DG-up-regulated DEGs were contained in clade I. CodeML found no significant positive selection on the basal node of clade Ia (*p* > 0.05), but RELAX suggested evolution under relaxed selection relative to the rest branches (*K* = 0.69; *p* = 0.003; Supplementary File 12). On the other subclades, no significant positive selection nor relaxed selection was found (Supplementary file 12). All DG-up-regulated DEGs were contained in clade I. It was hypothesized that duplication on the common lineage relates to plastid sequestration, and the PoB-specific duplication events contribute to long-term kleptoplasty.”